# Citrus Huanglongbing is a pathogen-triggered immune disease that can be mitigated with antioxidants and gibberellin

Wenxiu Ma[1,2], Zhiqian Pang[1,2], Xiaoen Huang [1,2], Jin Xu [1,2], Sheo Shankar Pandey [1,2], Jinyun Li[1,2], Diann S. Achor[1,2], Fernanda N. C. Vasconcelos [1], Connor Hendrich [1], Yixiao Huang[1], Wenting Wang[1], Donghwan Lee[1], Daniel Stanton[1] & Nian Wang [1✉]

Huanglongbing (HLB) is a devastating disease of citrus, caused by the phloem-colonizing bacterium *Candidatus* Liberibacter asiaticus (CLas). Here, we present evidence that HLB is an immune-mediated disease. We show that CLas infection of *Citrus sinensis* stimulates systemic and chronic immune responses in phloem tissue, including callose deposition, production of reactive oxygen species (ROS) such as $H_2O_2$, and induction of immunity-related genes. The infection also upregulates genes encoding ROS-producing NADPH oxidases, and downregulates antioxidant enzyme genes, supporting that CLas causes oxidative stress. CLas-triggered ROS production localizes in phloem-enriched bark tissue and is followed by systemic cell death of companion and sieve element cells. Inhibition of ROS levels in CLas-positive stems by NADPH oxidase inhibitor diphenyleneiodonium (DPI) indicates that NADPH oxidases contribute to CLas-triggered ROS production. To investigate potential treatments, we show that addition of the growth hormone gibberellin (known to have immunoregulatory activities) upregulates genes encoding $H_2O_2$-scavenging enzymes and downregulates NADPH oxidases. Furthermore, foliar spray of HLB-affected citrus with gibberellin or antioxidants (uric acid, rutin) reduces $H_2O_2$ concentrations and cell death in phloem tissues and reduces HLB symptoms. Thus, our results indicate that HLB is an immune-mediated disease that can be mitigated with antioxidants and gibberellin.

[1] Citrus Research and Education Center, Department of Microbiology and Cell Science, IFAS, University of Florida, Lake Alfred, FL, USA. [2] These authors contributed equally: Wenxiu Ma, Zhiqian Pang, Xiaoen Huang, Jin Xu, Sheo Shankar Pandey, Jinyun Li, Diann S. Achor. ✉email: nianwang@ufl.edu

Plants and animals utilize their immune system to fight pathogens. Both plants and animals have innate immunity, whereas animals also have adapted immunity. The plant innate immune system consists of pattern-triggered immunity (PTI), which is triggered by pathogen-associated molecular patterns (PAMPs) via cell surface-localized pattern-recognition receptors, and effector-triggered immunity (ETI), which is instigated by pathogen effector proteins via intracellular receptors called nucleotide-binding, leucine-rich repeat receptors (NLRs)[1–3]. However, some human diseases are mediated by the immune response itself, including autoimmune diseases such as inflammatory bowel disease, and non-autoimmune diseases, like asthma, sepsis induced by various microbes[4], and cryptococcal meningitis caused by *Cryptococcus neoformans*[5]. Host immune responses have been known to be an important factor in addition to microbial pathogenicity factors for human diseases caused by microbial pathogens, which has resulted in the proposal of a damage–response framework of microbial pathogenesis[6]. Immune-mediated diseases have not been reported in the Plantae Kingdom[7,8].

The damaging effect of plant diseases has been assumed to directly result from the impact of pathogenicity factors of the causal pathogens[9]. Common pathogenicity factors include effectors, toxins, cell wall degrading enzymes, and biofilm that are directly responsible for causing disease symptoms. For instance, the transcriptional activator-like effector PthA4 is responsible for the hypertrophy and hyperplasia symptoms of citrus canker caused by *Xanthomonas citri* subsp. citri[10]. Xylem blockage caused by biofilm of *Xylella fastidiosa* leads to the wilting of grapevine plants with Pierce's disease[11]. Citrus Huanglongbing (HLB, also known as citrus greening) is currently the most devastating citrus disease, causing billions of dollars of economic losses worldwide annually[12,13]. HLB presents an unprecedented challenge for the citrus industry despite some promising progress in research[14–16]. HLB is caused by the phloem-colonizing *Candidatus* Liberibacter asiaticus (CLas), *Ca.* L. americanus and *Ca.* L. africanus with CLas being the most prevalent in the world[17]. CLas is a biotroph, vectored by the Asian citrus psyllid (*Diaphorina citri*), and in the Rhizobiaceae family. Its ~1.23 Mb genome is significantly reduced compared to other free-living members of the Rhizobiaceae family, resulting from the reductive evolution within the nutrient-rich phloem of citrus and psyllid tissues[18].

Despite its economic importance, how *Ca.* Liberibacter damages infected citrus plants remains poorly understood, partly because HLB pathogens have not been cultured in artificial media. No pathogenicity factors have been confirmed to be responsible for the HLB symptoms of characteristic blotchy mottle on leaves, hardened leaves, small and upright leaves, leaves showing zinc or manganese deficiency, corky veins, twig dieback, stunted growth of seedlings, thin canopy, small and lopsided fruit and root decay[17].

In this work, we present evidence that citrus HLB is an immune-mediated disease. This hypothesis explains most HLB phenomena, is consistent with the genetic information of *Ca.* Liberibacter spp., and provides guidance regarding HLB management.

## Results

### CLas does not contain pathogenicity factors that directly cause HLB symptoms.
We conducted a comprehensive analysis of CLas proteins and did not identify any homologs with known pathogenicity factors that are directly responsible for causing plant disease symptoms. *Ca.* Liberibacter spp. do not contain type II, III, or IV secretion systems that secrete such pathogenicity factors. The pathogenicity genes of closely related *Agrobacterium*

and *Rhizobium* pathogens[19,20] were not identified in CLas. To test whether CLas contains any novel pathogenicity factors responsible for causing HLB symptoms, 47 putative virulence factors including serralysin and hemolysin (substrates of type I secretion system) and proteins containing Sec secretion signals (Table S1)[21] were overexpressed in *Arabidopsis thaliana*, *Nicotiana tabacum* or *Citrus paradisi*. These virulence factors refer to genes that contribute to bacterial growth in plants but are not directly responsible for disease symptoms, and genes that contribute to virulence in non-plant hosts, such as serralysin. None of the overexpressed CLas proteins caused HLB-like symptoms, consistent with the bioinformatic analyses that CLas does not contain pathogenicity factors that directly cause HLB symptoms. Intriguingly, it was reported that CLas triggers immune responses[22–26], which we hypothesize is responsible for causing the devastating damages of HLB, similar to immune-mediated diseases of human.

### CLas infection triggers immune response and cell death in the phloem tissue.
Next, we tested if and how CLas triggers an immune response and cell death. Newly emerged citrus flush from HLB-positive citrus trees is initially free of CLas. We trimmed HLB-positive and healthy 2-year-old *C. sinensis* "Valencia" trees in a greenhouse to trigger young flush and conducted dynamic analyses of the relationship between CLas infection, immune response, cell death, and HLB symptom development. CLas was detected in young leaves of HLB-positive trees at ~15 days post-bud initiation based on quantitative PCR (qPCR) (Fig. 1). $H_2O_2$ (an indicator of reactive oxygen species (ROS)) content in CLas-positive flush was significantly higher than that of healthy plants at 15- and 21-day post-bud initiation (Fig. 1a). Significantly more callose deposition, an indicator of immune response[27], was observed in CLas-positive flush than that of the healthy plants starting at 18 days post-bud initiation and thereafter (Fig. 1b). The observation that CLas triggers ROS production earlier than callose deposition is consistent with the fact that ROS positively regulate callose deposition[28]. On the other hand, significantly more starch accumulation, which was suggested to cause disruption of chloroplast inner grana structure and contribute to the yellowing symptoms[29], was observed in CLas-positive samples than in healthy samples beginning 18 days after bud initiation (Fig. 1c). Symptoms began to appear at ~40 days after bud initiation. It appears that CLas infection triggered plant immune responses, such as ROS (e.g., $H_2O_2$) production and callose deposition, that in turn, incited symptom development.

To better understand the nature of immune responses induced by CLas, we conducted temporal expression analyses of immune marker genes (*PR1*, *PR2*, *PR3*, and *PR5*) in young leaves at 15-, 18-, 21-, 24-, 27-, and 60-days post-bud initiation for the CLas infected and healthy *C. sinensis* plants. PR genes were consistently induced by CLas despite some fluctuations between 15- and 60-day-post-bud initiation (Fig. S1).

We observed cell death of sieve element and companion cells via transmission electron microscopy (TEM) of asymptomatic young leaves of HLB-positive *C. sinensis* "Valencia" trees (Fig. 2a–d), indicating cell death of phloem tissues occurs prior to the appearance of HLB symptoms. CLas was observed in phloem tissue with intact sieve element and companion cells, but not in sieve element cells undergoing the cell death process (Fig. 2c–e). It was noteworthy that some sieve element and companion cells underwent cell death while others remained intact in the same field (Fig. 2d), explaining the reduced function of the phloem, rather than its complete loss of function. This is consistent with the observation that CLas moves primarily

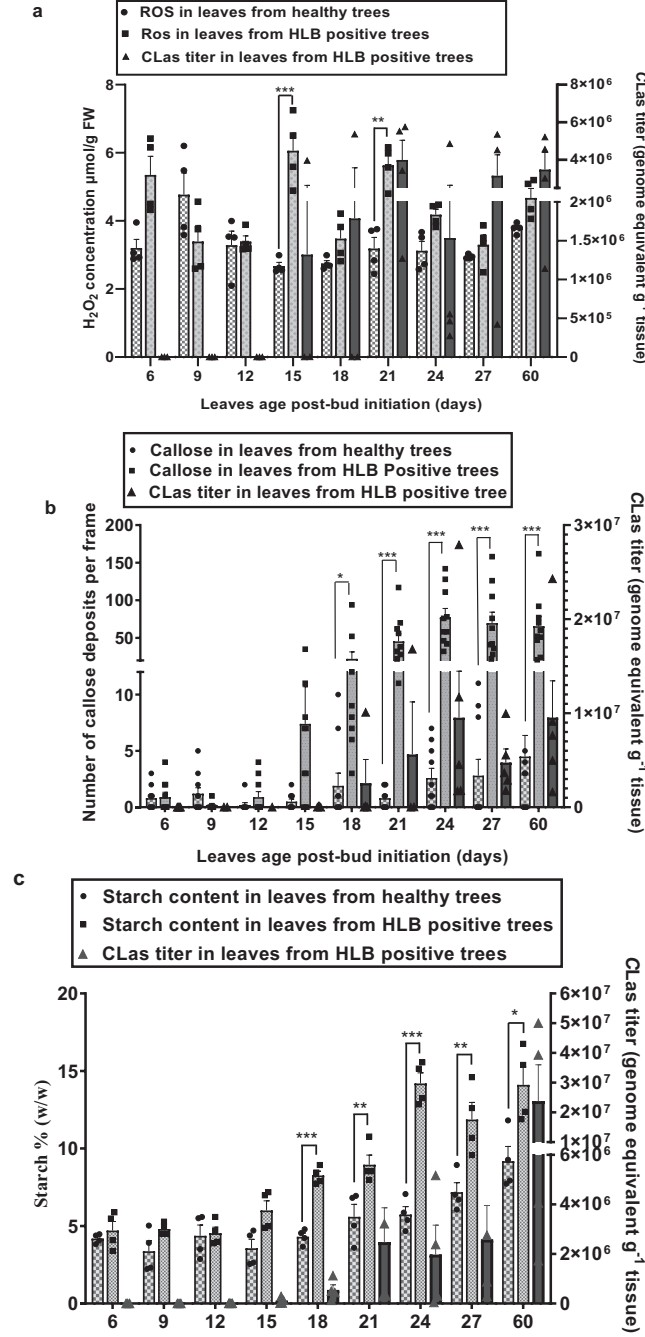

**Fig. 1 CLas infection causes ROS production, callose deposition, and starch accumulation in young citrus leaves. a** Quantification of $H_2O_2$ in young leaves of 2-year-old HLB positive and healthy *C. sinensis* "Valencia" trees. Data shown are mean ± SEM, $n = 4$. **b** Quantification of callose depositions of young leaves by staining with aniline blue and observed under an epifluorescence microscope. Data shown are mean ± SEM, $n = 10$. Callose depositions were counted from ten leaves from five trees. **c** Starch content in young flush. Data shown are mean ± SEM, $n = 4$. HLB positive and healthy 2-year-old *C. sinensis* "Valencia" trees were trimmed to trigger young flush in the greenhouse. qPCR was conducted to quantify CLas titers in newly emerged leaves at different time points after bud initiation. Experiments were repeated twice with similar results. *, **, and *** indicate $p$ value < 0.05, 0.01, and 0.001, respectively, compared with healthy young leaves and calculated using two-tailed Student's *t* test. Source data are provided as a Source Data file.

vertically, but not laterally during infection[30]. TEM observation of CLas-infected leaves confirmed that cell death was limited to sieve element and companion cells[31], but not in surrounding parenchyma cells (Fig. 2c).

We confirmed cell death in mature *C. sinensis* leaves exhibiting various symptoms using trypan-blue staining. No cell death was observed in healthy leaves collected from CLas-free plants. Cell death was observed in CLas-positive leaves and was more severe in leaves with severe symptoms (Fig. 3a), congruent with the TEM observation of the death of sieve element and companion cells in the midribs of CLas-infected mature leaves and CLas-positive stem tissues (Figs. S2, S3). Higher incidence of cell death was observed in leaves with higher CLas titers, suggesting CLas infection is responsible for cell death in the phloem tissue (Figs. S2, S3). TEM demonstrated that death of both companion cells and sieve element cells (Fig. 2c, d) or death of companion cells alone (Figs. 2b, S3c), but not death of sieve element cells alone occurred, indicating that death of companion cells occurred before that of sieve elements. We observed significantly higher $H_2O_2$ concentrations in CLas-infected mature leaves than CLas-free leaves (Fig. 3b), consistent with a previous study by Pitino et al.[26]. In addition, significantly higher $H_2O_2$ concentrations were detected in the exudates of phloem-enriched tissues of CLas-positive stems than in stems of CLas-free trees (Fig. 3c).

Cell death is usually accompanied by ion leakage. Surprisingly, no difference in ion leakage between leaf blades or midribs of healthy, asymptomatic, mildly symptomatic, and severely symptomatic leaves was observed (Fig. S4), contrary to the results of TEM and trypan-blue staining (Figs. 2, 3a, S2). However, ion concentrations in exudates extracted from phloem-enriched bark tissues of CLas-infected samples were significantly higher than that of healthy samples (Fig. 3d), consistent with a model in which cell death occurs in phloem tissues, but not in surrounding parenchyma (Fig. 2c). Any change in ion leakage in the leaf blades and midribs of CLas-infected samples (Fig. S4) is probably masked by parenchyma cells because companion and sieve element cells make up only ~1% of the total cell population[32].

Next, we used callose deposition as an indicator to investigate the localization of the immune response of citrus leaves in response to CLas infection. For this test, we investigated callose deposition in different sections of asymptomatic and symptomatic leaves of HLB-positive *C. sinensis* trees (Figs. 4a–g, S5). Callose deposition in the petiole, midrib, and lamina of asymptomatic leaves was significantly lower than that in symptomatic leaves (Figs. 4a–g, S5). No callose deposition was observed in the CLas-free lamina of asymptomatic leaves (Figs. 4c, S5). Together, the correlation between callose deposition and CLas titers suggests that CLas is responsible for inducing callose deposition (Figs. 4a–f, S5) as observed in the past[24]. Callose deposition was observed only in phloem tissues as observed previously[33], but not in mesophyll cells. In contrast, pathogens infecting the apoplast, such as *Xanthomonas*, induce callose that is deposited between the plasma membrane and the cell wall in the vicinity of the pathogen[34,35]. Hence, CLas induces a systemic immune response in phloem tissues following its systemic infection of the tree.

To further verify that CLas induces immune responses in phloem tissues, we monitored ROS formation and localization in phloem-enriched bark tissues using the fluorescent probe 2′,7′-dichlorodihydrofluorescein diacetate ($H_2DCFDA$) and confocal laser microscopy. $H_2DCFDA$ is a commonly used cell-permeable probe for measuring cellular $H_2O_2$[36]. Significantly higher levels of $H_2O_2$ were detected in the phloem-enriched bark tissues of CLas-infected citrus plants than that of CLas-free plants (Fig. 4h–j).

Next, we aimed to further establish the causal relationship between CLas infection and ROS induction and cell death in

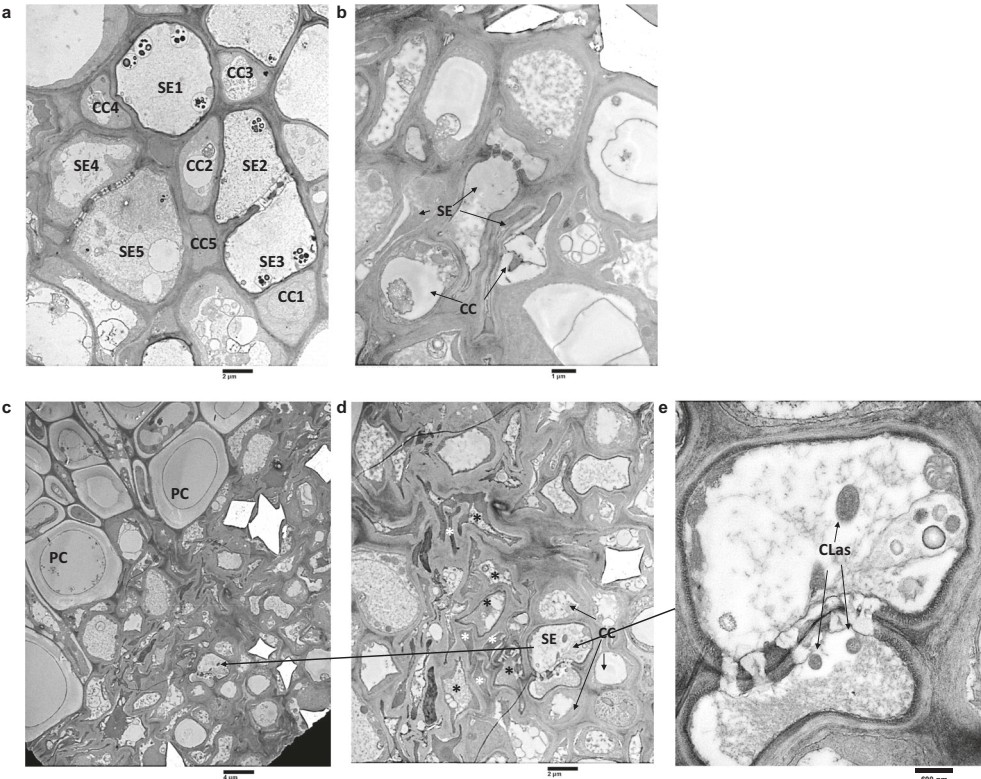

**Fig. 2 Transmission Electron Microscopy (TEM) analysis of asymptomatic young leaves of CLas-infected *Citrus sinensis*. a** Heathy leaf of *C. sinensis* "Valencia". CC (companion cells): typical healthy companion cells with large central nucleus, dense cytoplasm, numerous mitochondria. SE (sieve element 1–3): typical sieve elements showing parietal "s" plastids, centrally distributed p protein, and lateral sieve plate showing minimum callose. SE (4–5): developing sieve elements showing plastids with no starch, intact central vacuole, and non-dispersed phloem protein. **b** Asymptomatic young leaf of CLas-infected *C. sinensis*. CC undergoing cell death. **c**, **d**, **e** Asymptomatic young leaves showing cell death of sieve element cells and companion cells, but not parenchyma cells (PC). Black * indicates cell death of sieve element cells. White * indicates cell death of companion cells. **d** Enlarged section of C showing both intact sieve element and companion cells as well as sieve element and companion cells undergoing cell death. **e** Enlarged section of D showing intact sieve element cells containing CLas. Experiments were repeated more than three times with similar results.

phloem tissues. CLas-positive 5-year-old *C. sinensis* trees were treated with streptomycin via trunk injection to kill the pathogen[37]. At 7 days after treatment, streptomycin significantly reduced CLas titers, $H_2O_2$ content, and ion leakage in the phloem tissue (Fig. 4k–l). Collectively, this establishes the causative relationship between CLas infection and ROS induction and cell death in phloem tissues.

**HLB-mediated cell death is triggered by ROS.** Cell death can be instigated by ROS[38]. At high concentrations, ROS trigger necrotic cell death, but also induce programmed cell death at lower concentrations[39]. We thus analyzed $H_2O_2$ contents in *C. sinensis* leaves infected by CLas. The $H_2O_2$ concentration in young CLas-infected leaves was ~6 µmol g$^{-1}$ FW, but reached 10–15 µmol g$^{-1}$ FW in mature leaves (Fig. 3b). Moreover, this method probably underestimated the $H_2O_2$ concentration in the phloem tissue because it could not differentiate levels in phloem cells, where $H_2O_2$ concentrates (Fig. 4h–j), from that in the more abundant parenchyma cells. Intriguingly, *Xanthomonas citri* subsp. citri, another bacterial pathogen of citrus, triggers $H_2O_2$ production by 2 days after inoculation of kumquat (*Citrus japonica*, syn: *Fortunella crassifolia*) which peaks (9.86 µmol g$^{-1}$ FW) 8 days after inoculation, eventually leading to cell death[40,41]. *X. citri* subsp. citri-induced cell death is a slow process, occurring 6–8 days after inoculation[41]. Hence, we conjectured that ROS induced by CLas can reach a threshold necessary to trigger death of companion and sieve element cells in mature leaves. ROS production triggered by CLas is distinct from that triggered by incompatible

pathogens, which is typified by a biphasic oxidative burst[42]. Instead, high levels of ROS were consistently detected in both young leaves during early stages of infection (Fig. 1a) as well as in mature infected leaves and stems (Fig. 3b, c), probably triggered by CLas colonizing and multiplying in the previously unoccupied phloem tissue as well as that triggered by damage-associated molecular patterns resulting from dying companion and sieve element cells.

$H_2O_2$ concentrations in the phloem-enriched bark tissues from symptomatic (1.80 ± 0.13 mmol/L) branches were much higher than that (0.59 ± 0.01 mmol/L) of healthy trees (Fig. 3c). $H_2O_2$ induces necrosis of immortalized rat embryo fibroblasts at a concentration of 700 µmol/L[43]. $H_2O_2$ at a concentration of 1.8 mmol/L but not 0.6 mmol/L or lower induced cell death of *C. sinensis* protoplast cells (Figs. 5a, b, S6). Similar results were observed for suspension culture cells (Fig. S7). Addition of uric acid (0.2 mM), a ROS scavenger, reduced both $H_2O_2$ concentrations (Fig. 5c) and cell death (Fig. 5a, b), indicating that $H_2O_2$ induced by CLas is sufficient to kill phloem tissue. It is important to note that the growth of plants (e.g., *Arabidopsis*) is inhibited by 1 mM $H_2O_2$[44], partly explaining the growth stunting phenotype of CLas-infected young citrus trees.

In addition to $H_2O_2$, ROS induced by pathogens include hydroxyl radicals, superoxide anions, and singlet oxygen. To further corroborate that HLB caused cell death is instigated by ROS, we sprayed HLB-positive *C. sinensis* 'Valencia' trees weekly with the antioxidants uric acid (1.8 mM) and rutin (0.6 mM). After treatment for 6 weeks, ROS levels in phloem-enriched bark

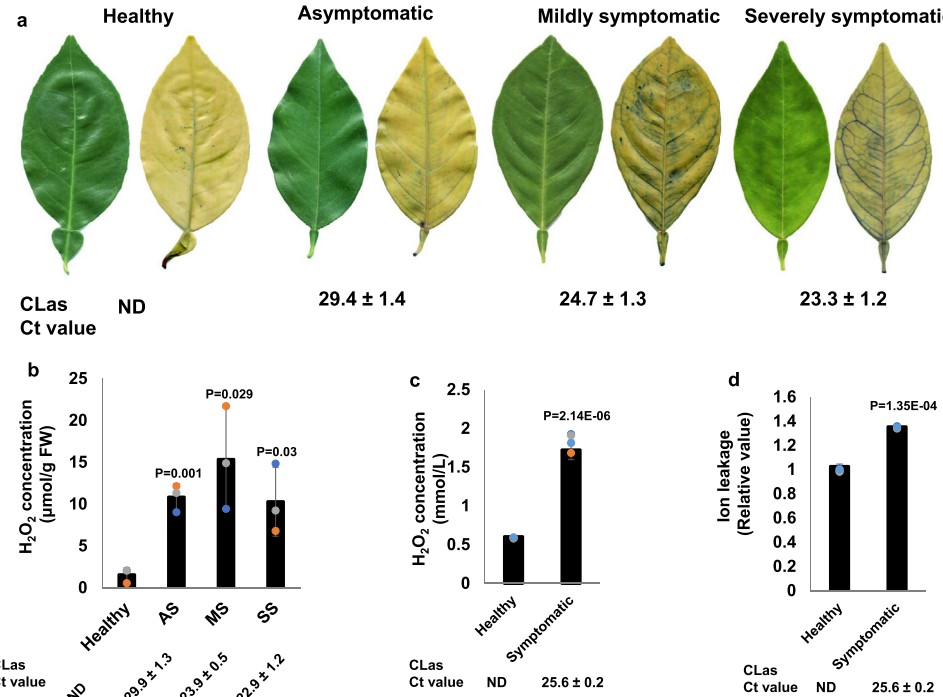

**Fig. 3 *Candidatus* Liberibacter asiaticus (CLas) induces ROS production and cell death in phloem tissue of *C. sinensis*. a** Trypan-blue staining assay to detect cell death in leaves of CLas-infected *C. sinensis* "Valencia" trees. **b** Determination of $H_2O_2$ concentration in CLas negative or positive leaves showing different symptoms. Mean and SD ($n = 3$) are shown. **c, d** $H_2O_2$ (**c**) and ion leakage (**d**) assays of exudates extracted from phloem-enriched bark tissues. Data are presented as mean values ± SD ($n = 4$). Statistical differences were analyzed using two-tailed Student's *t* test. ND non-detected. For (**a, b**): AS asymptomatic. MS Mildly symptomatic. SS Severely symptomatic. AS, MS, and SS were collected from HLB-positive trees. For (**c, d**), Healthy stems of spring sprouts were collected from healthy branches of CLas-negative plants. Symptomatic stems of spring sprouts were collected from branches with HLB symptomatic leaves of CLas-positive plants. Each experiment contains three to four biological replicates. The experiments were repeated at least twice with similar results. CLas titers were determined by qPCR as shown by Ct values with lower Ct values indicating higher bacterial titers. Source data are provided as a Source Data File.

tissues were much lower than control plants (Fig. 5c), and the uric acid and rutin treatments reduced cell death (Fig. 5d). Taken together, CLas infection of citrus phloem tissue induces ROS production, which subsequently causes cell death of phloem tissue.

**CLas infection significantly altered expression of pathways related to oxidative stress and immune responses.** Next, we investigated 20 previously published gene expression profiles of *C. sinensis* response to CLas infection that studied different tissues (leaf, stem, and fruit), different environments (greenhouse or groves), and different infection stages (Table S2). Analyses of differentially expressed genes (DEGs) clearly demonstrated that the expression of genes related to ROS and immune response are significantly altered by CLas infection (Table S3). The combined analysis showed an overall downregulation of antioxidant enzymes and upregulation of transmembrane localized NADPH oxidases, known as RBOHs, explaining the oxidative stress response in response to CLas infection (Fig. S8). Critically, expression of respiratory burst oxidative homolog D (RBOHD) gene, which encodes NADPH oxidase implicated in the generation of ROS during defense responses, was induced by CLas infection in several studies. RBOHD is primarily responsible for ROS produced upon PAMP recognition and is required for cell death that is initiated after pathogen detection[45]. In addition, RBOHB and RBOHF have also been reported to be involved in ROS production in response to pathogen infection[45,46]. qRT-PCR assays revealed that both *RBOHB* and *RBOHD* were induced by CLas infection of leaf samples collected in both field and greenhouse conditions, whereas *RBOHF* was induced only under field

conditions (Fig. S9). Intriguingly, ROS levels in CLas-positive stems were reduced by NADPH oxidase inhibitor diphenyleneiodonium (DPI) (Fig. S10), supporting the notion that RBOHs contribute to the ROS accumulation triggered by CLas, which probably also results from contributions from other components such as peroxidases.

Taken together, our analysis implies that CLas infection causes oxidative stress to citrus in most conditions, consistent with the $H_2O_2$ production it triggers (Figs. 1a, 3b, c, 4h–k, 5c). Intriguingly, the involvement of oxidative stress in HLB disease has been suggested by multiple previous studies[26,47,48]. Comparative transcriptome analysis revealed complex expression changes associated with the immune response pathways in response to CLas infection (Fig. S11). Approximately 66 NLR genes showed overall induction by CLas in many studies (Fig. S11B). Transcriptome analyses of sweet orange response to CLas infection therefore support our hypothesis that HLB is an immune-mediated disease that results from ROS induced cell death of phloem tissue triggered by the pathogen.

**Suppressing ROS-mediated cell death mitigates HLB symptoms.** Antioxidants, and immunoregulators are commonly used to treat human immune-mediated diseases by halting or reducing ROS-mediated cell death[49–51]. Correspondingly, we tested whether growth hormone gibberellin (GA), and antioxidants (uric acid and rutin) could mitigate the ROS-mediated cell death triggered by CLas infection, and thus block or reduce HLB symptoms. The plant growth hormone GA was selected because it is known to modulate PAMP-triggered immunity and PAMP-

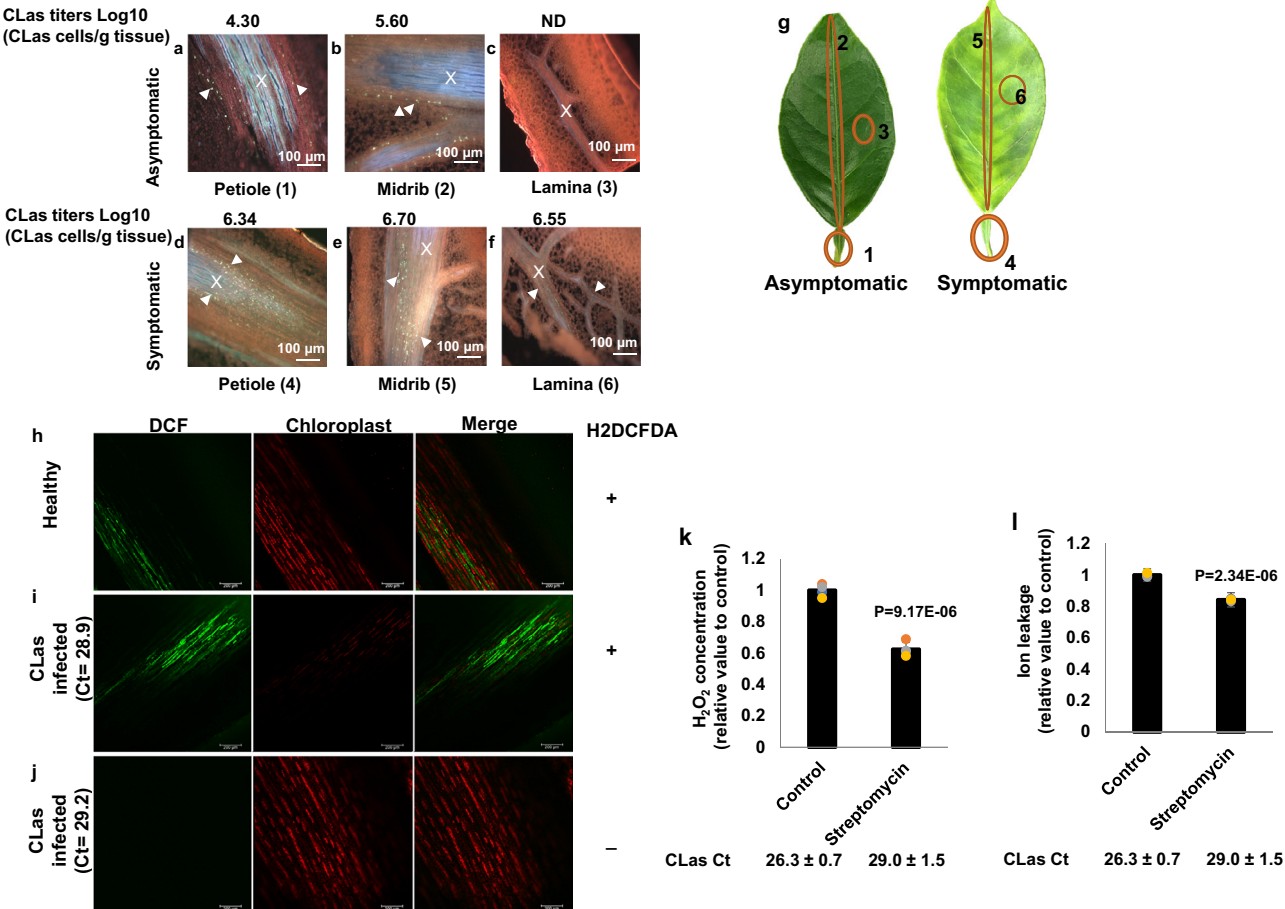

**Fig. 4 CLas infection induces callose deposition, H₂O₂ production and cell death in the phloem tissue of _C. sinensis_. a–f** _C. sinensis_ "Hamlin" leaf samples were fixed with FAA solution overnight, sectioned and stained with 0.005% aniline blue solution prior to analysis. Xylem is marked with an X, and callose deposition is indicated with arrowheads. Asymptomatic samples: (**a**) Petiole, (**b**) Midrib, (**c**) Lamina; Symptomatic samples: (**d**) Petiole, (**e**) Midrib, and (**f**) Lamina. Pictures are representatives of 12 replicates. ND Non-detected. **g** schematic representation of samples used for callose deposition assays (**a**-**f**). CLas titers for each section were quantified by qPCR. **h–j** H₂O₂ production in the phloem tissue. Healthy (**h**) and CLas-infected (**i**) _C. sinensis_ "Valencia" bark tissues visualized with 2′,7′-dichlorodihydrofluorescein diacetate (H₂DCFDA) under a confocal laser microscope. CLas infected (**j**) _C. sinensis_ "Valencia" bark without H2DCFDA was used as a control. "+" indicates with H₂DCFDA treatment. "−" indicates without H₂DCFDA treatment. **k–l** Effect of killing CLas with streptomycin on H₂O₂ concentration (**k**) and cell death (**l**) in the phloem tissue. CLas-positive 5-year-old sweet orange trees were trunk-injected with streptomycin. Non-treatment was used as the negative control. The tests were conducted 7 days after trunk injection of streptomycin. Four biological replicates were used and mean values ± SD (_n_ = 4) are shown. Two-tailed Student's _t_ test was used for statistical analysis. Experiments were repeated at least two times with similar results and representative results are shown. Source data are provided as a Source Data File.

induced plant growth inhibition[52]. Both uric acid and rutin were also assessed since they are well-known ROS scavengers[53,54].

Foliar sprays of HLB-positive _C. sinensis_ trees with GA at both 5 mg/L and 25 mg/L reduced tissue H₂O₂ levels and ion leakage (Fig. 6a, b). GA (5 mg/L) also suppressed death of _C. sinensis_ protoplast cells treated with 1.8 mM H₂O₂ (Fig. 6c, d). RNA-seq analyses of GA treated vs. non-GA treated _C. sinensis_ protoplast cells in the presence of 1.8 mM H₂O₂ demonstrated that GA induced the expression of genes encoding H₂O₂ scavenging enzymes catalases, ascorbate peroxidases, and glutathione peroxidases. GA also inhibited the expression of _RBOHD_ (Fig. 7, Supplementary Data 1). GA treatment therefore clearly alleviates the oxidative stress caused by H₂O₂ in citrus.

Six weeks after initiation of foliar sprays, treated plants exhibited reduced HLB symptoms (i.e., less blotchy mottle) compared with that of plants before treatment, whereas the blotchy mottling symptoms of plants treated only with water became more severe in this period (Fig. S12). In addition, GA promoted plant growth as indicated by new leaf flushes that were not observed on water-treated plants (Fig. S12). GA is registered

for use on citrus in the US, enabling its use in large-scale field trials of its efficacy for control of HLB. Foliar spray of _C. sinensis_ with GA significantly reduced HLB disease symptoms 8 months after application in such large-scale trials (Figs. 6e–g, S13). The treated trees appeared much healthier than the non-treated control trees even though all were infected (Fig. 6g). Because symptomatic leaves demonstrated significantly more cell death than asymptomatic leaves based on trypan-blue staining (Fig. 3a) and TEM observation (Fig. S2), we used the proportion of symptomatic leaves on a tree as an indicator of cell death caused by HLB. GA treatment significantly reduced the incidence of symptomatic leaves (i.e., leaves showing blotchy mottle, yellowing, and nutrient deficiency) (Figs. 6e–g, S13), suggesting reduced death of sieve element and companion cells in treated leaves. In addition, foliar sprays of GA on HLB-positive 6-year-old _C. sinensis_ var. 'Valencia' and var. 'Vernia' significantly promoted plant growth as measured by tree height, trunk diameter and canopy volume when assessed 8 months after application (Fig. S14). We presume GA reduces HLB symptoms via its direct effect on both mitigating ROS (Figs. 6a, c, d, 7) and promoting

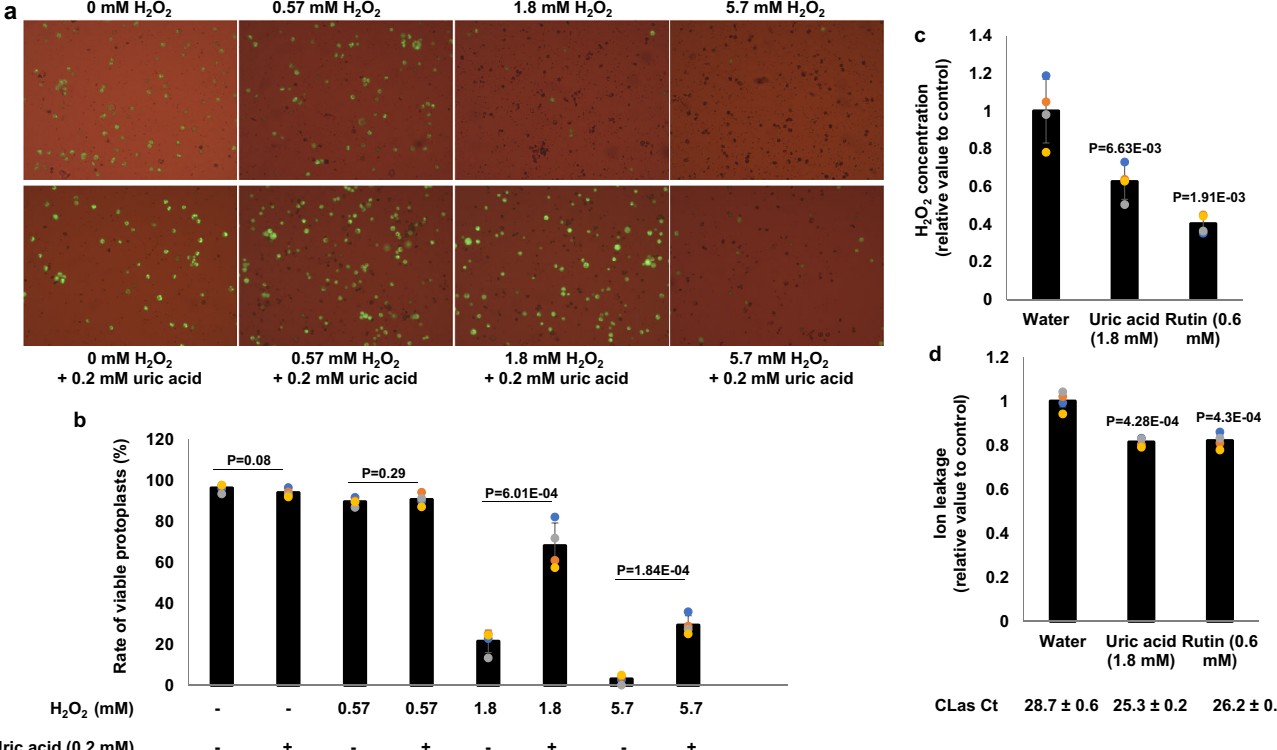

**Fig. 5 ROS are responsible for cell death of phloem tissues of CLas-infected citrus. a**, **b** $H_2O_2$ kills protoplast cells of *C. sinensis*. **a** Freshly prepared protoplast cells of *C. sinensis* were treated with different concentrations of $H_2O_2$ with or without the antioxidant uric acid (0.2 mM) for 24 h and tested for viability via fluorescein diacetate (FDA) staining. **b** Quantification of viable protoplast cells in different treatments of (**a**). **c**, **d** HLB-positive plants were treated with antioxidants via foliar spay weekly for 6 weeks. The exudates extracted from phloem-enriched bark tissues were used for detection of $H_2O_2$ and ion leakage. **d** Uric acid (1.8 mM) and rutin (0.6 mM) reduce $H_2O_2$ concentration triggered by CLas in the phloem tissue. **d** Uric acid (1.8 mM) and rutin (0.6 mM) reduce ion leakages in phloem tissues infected by CLas. Statistical differences were analyzed using one way ANOVA with Bonferroni Correction ($P < 0.05$). Each experiment contains four biological replicates. Different letters above the columns indicate statistical differences ($P < 0.05$). Ct values of CLas of the tested samples were indicated. Source data are provided as a Source Data File.

plant growth (Fig. S14), thus alleviating the growth inhibition caused by CLas.

To determine whether mitigating ROS can directly halt or reduce HLB symptoms, we sprayed HLB-positive symptomatic *C. sinensis* trees weekly with the antioxidants uric acid (1.8 mM) and rutin (0.6 mM). Remarkably, 6 weeks after initial treatment, both antioxidants significantly reduced HLB symptoms compared to that observed before treatment, whereas the plants treated with water became more symptomatic in the same duration (Fig. S12). Taken together, our data suggest that suppression of ROS-mediated cell death can mitigate HLB symptoms. Consequently, we have established the causal relationship that CLas triggers ROS production in the phloem tissue, which subsequently causes cell death of phloem tissue, leading to HLB symptoms.

## Discussion
In this study, we provide evidence that citrus HLB is an immune-mediated disease. CLas induces a systemic chronic immune response, mimicking systemic chronic inflammation diseases of humans[55]. Systemic chronic inflammation diseases may result from collateral damage to tissues and organs over time by oxidative stress[56]. ROS concentrations triggered by CLas infection are above the threshold needed to induce cell death, probably resulting from the combined effect of programmed cell death induced at low ROS concentrations and necrotic cell death stimulated at high ROS concentrations. In addition, ROS positively regulate callose deposition[28,57] and inhibit plant growth including roots[58,59]. Persistent induction of ROS by systemic CLas infection

leads to systemic cell death of phloem tissue and other effects owing to diverse roles of ROS, which subsequently affect phloem function, hormone synthesis and transport, metabolic transport, and rerouting energy to immune responses rather than to growth. This model can explain most HLB phenomena. For instance, phloem dysfunction resulting from the death of companion and sieve element cells can lead to starch accumulation, and the resultant blotchy mottle symptoms. Hardened leaves likely result from the action of ROS which are known to directly cause strengthening of host cell walls[42]. Cell death of the phloem tissue, reduced transport of photosynthates, and ROS inhibition of root growth may be responsible for root decay. Stunted growth probably results from the direct effect of ROS and reduced transport of carbohydrates and hormones. The detailed molecular mechanism of how CLas activates immune responses remains unknown. We anticipate that cytoplasmic receptors, such as nucleotide-binding leucine-rich repeat (NLR) proteins are primarily responsible for intracellular detection of CLas through recognition of PAMPs inside companion and sieve element cells. Although they have not been previously known for plants, immune-mediated diseases are likely prevalent in the Kingdom Plantae, particularly for diseases caused by phloem-colonizing bacteria (e.g., *Ca.* Liberibacter, *Spiroplasma*, and *Ca.* Phytoplasma), viruses, and fungi as well as non-phloem-colonizing pathogens.

The finding of citrus HLB as an immune-mediated plant disease helps guide the battle against this notorious disease. It seems likely that horticultural approaches that suppress oxidative stress

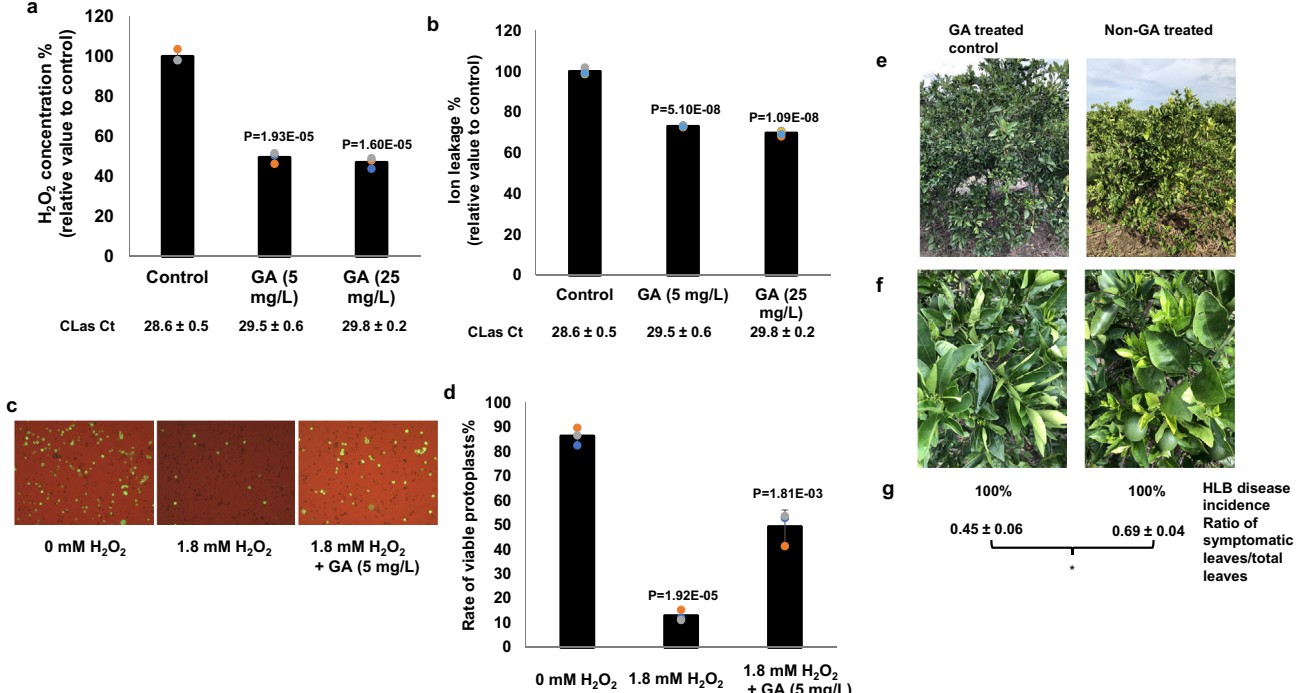

**Fig. 6 The immunoregulator gibberellin (GA) suppresses HLB development. a, b** GA suppresses ROS-mediated cell death. HLB-positive *C. sinensis* trees were treated with GA via foliar spray weekly for 6 weeks. The exudates extracted from phloem-enriched bark tissues were used for detection of $H_2O_2$ (**a**) and ion leakage (**b**). Ct values of CLas of the tested samples were indicated. **c, d** GA suppresses cell death of *C. sinensis* protoplast cells. Freshly prepared protoplast cells of *C. sinensis* were treated with $H_2O_2$ with or without GA (5 mg/L) for 24 h and tested for viability via fluorescein diacetate (FDA) staining. Statistical differences (**a, b, d**) were analyzed using two-tailed Student's *t* test. Mean ± SD (**a**, $n = 3$; **b**, $n = 4$; **d**, $n = 3$) are shown. *P* valued are shown above each columb. **e–g** Foliar spray of GA suppresses HLB symptoms. *C. sinensis* "Valencia" blocks were treated with GA (1247 ppm) in November 2020. Nearby blocks of *C. sinensis* "Valencia" that were not treated with GA were used as negative controls. Symptoms, HLB disease incidence and ratio of symptomatic leaves/total leaves were investigated in June 2021. **e** Representative whole trees. **f** Representative sections. **g** HLB disease incidence and ratio of symptomatic leaves vs. total leaves in different treatments. * indicates *P* value <0.05 based on Student's *t* test. Pictures were taken at the same day in June 2021. Source data are provided as a Source Data File.

can alleviate the immune-mediated damage caused by CLas in HLB endemic citrus production areas. These approaches include optimized usage of plant growth hormones, such as GA and brassinosteroids[60], and antioxidant treatments. Even though we did not test the effect of nutritional modulation of immune function on HLB in this study, citrus growers in Florida have observed that modulation of macronutrients (N, P, and K) and micronutrients (e.g., B, Cu, Fe, Mn, Mo, Ni, Se and Zn) can reduce HLB symptoms. This is consistent with the observation that a deficiency of macronutrients can lead to oxidative stress[61], whereas micronutrients (B, Cu, Fe, Mn, Mo, Ni, Se and Zn) at low concentrations can activate antioxidative enzymes[62]. Antioxidants (e.g., uric acid), growth hormones (e.g., GA) and nutritional modulation (e.g., micronutrients) could be used to directly alleviate oxidative stress to reduce cell death of the phloem tissue to mitigate HLB symptoms. Moreover, growth hormones and nutrients, by promoting new growth, decrease the proportion of dead cells in phloem tissue, further mitigating HLB symptoms. The horticultural measures used to mitigate ROS and cell death are unable to reduce or eliminate CLas inoculum, thus are not recommended for citrus production areas with low HLB incidence. For those areas, region-wide comprehensive implementation of roguing infected trees, tree replacement, and insecticide applications has been shown to successfully control citrus HLB[17,63]. Genetic improvements that enhance plant tolerance of oxidative stress, prevent overproduction of ROS, or evade recognition of CLas may generate HLB resistant/tolerant citrus varieties. In particular, enhanced companion cell- or

phloem-specific overexpression of antioxidant enzymes (such as superoxide dismutase, catalases, glutathione peroxidases, ascorbate peroxidase, and glutathione reductase) could be achieved using CRISPR-mediated gene editing, transgenic, or cisgenic approaches or citrus tristeza virus vectors). Additionally, gene editing of the NLRs responsible for recognizing CLas might also render citrus tolerant to HLB. It is plausible that some of the HLB-tolerant citrus genotypes might not recognize PAMPs or proteins of CLas or the recognition is too weak to reach the ROS threshold to cause cell death, thus avoiding the immune-mediated damages. On the other hand, some HLB-tolerant citrus genotypes might be superior in antioxidant mechanism or phloem regeneration. Intriguingly, it has been reported that HLB-tolerant varieties contain higher levels of antioxidants than more susceptible varieties[64]. "Sugar Belle" mandarin, which is HLB-tolerant, has more vigorous phloem regeneration than the HLB-susceptible *C. sinensis* varieties[15]. In summary, citrus HLB is an immune-mediated disease and mitigating ROS via antioxidant mechanisms and promoting new growth both can reduce cell death of phloem tissues, thus controlling HLB.

## Methods

**Transgenic expression analysis of CLas proteins containing Sec secretion signals and other putative virulence factors.** Transgenic expression of CLas genes was conducted[65]. For the citrus transformation, CLas genes were amplified without signal peptide sequence and cloned into the binary vector RCsVMV-erGFP-pCAMBIA-1380N-35S-BXKES-3xHA, which has a C-terminal 3×HA tag, to generate the CLas gene overexpression vectors. The resulting binary vectors were transferred into *Agrobacterium tumefaciens* strain EHA105 for citrus

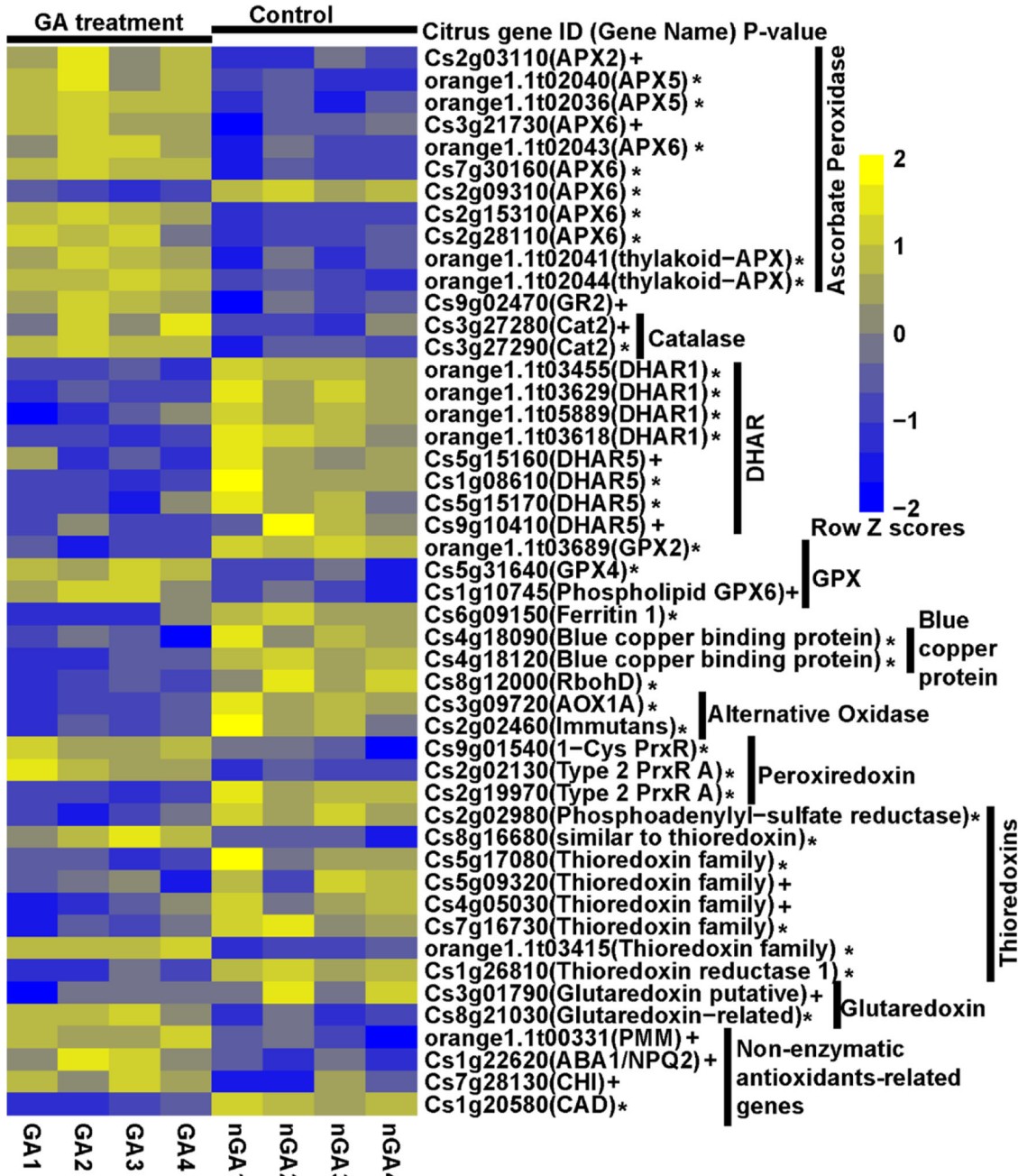

**Fig. 7 The expression profiling of ROS related genes between GA and non-GA (nGA) treated *Citrus sinensis* protoplast cells in the presence of 1.8 mM H₂O₂.** The genes include ROS related genes encoding ascorbate peroxidase, glutathione reductase, catalase, dehydroascorbate reductase (DHAR), glutathione peroxidase (GPX), ferritin and blue copper protein, NADPH oxidase, alternative oxidase, peroxiredoxin, thioredoxins, glutaredoxin, and non-enzymatic antioxidants-related genes. The *P* value was determined using the DESeq2 method and corrected using the BH (Benjamini-Hochberg) method. The asterisk denotes *P* value < 0.01; the plus sign denotes *P* value < 0.05. The gene expression value was calculated using the RPKM method. Scale indicates expression value of each gene after row normalization by removing the mean (centering) and dividing by the standard deviation (scaling).

transformation. The empty vector (EV) was used in citrus transformation as a negative control. *Agrobacterium*-mediated transformation of epicotyl segments of Duncan grapefruit (*Citrus paradisi*) was carried out to generate transgenic citrus plants[66]. Transgenic lines showing kanamycin-resistance and erGFP-specific fluorescence were selected and then micro-grafted in vitro onto 1-month-old Carrizo citrange rootstock seedlings. After 1 month of growth in vitro, the grafted shoots were potted into a peat-based commercial potting medium and acclimated under greenhouse conditions for the phenotype evaluation. Transgenic plants were confirmed by PCR, qRT-PCR at the RNA level, or western blot using HA Tag Antibodies (Cat. #11867423001, Roche).

For the tobacco transformations, *Agrobacterium*-mediated transformation of leaf discs of *Nicotiana tabacum* was carried out to generate the transgenic tobacco[67]. *A. tumefaciens* strain EHA105 containing the vectors was used for transformation.

Transgenic positive shoots showing kanamycin-resistance and erGFP-specific fluorescence were selected and transferred to the rooting medium. Evaluation of the transgenic *N. tabacum* was conducted in a growth chamber. Transgenic plants were confirmed by PCR, qRT-PCR at the RNA level, or western blot using HA Tag Antibodies (Cat. #11867423001, Roche).

For the gene overexpression in *Arabidopsis thaliana*. CLas genes without signal peptides were PCR amplified and cloned into the binary vector pCambia1380-35S-EYFP, which has a C-terminal EYFP protein tag, and transferred into *A. tumefaciens* strain GV2260. *Agrobacterium*-mediated floral dip method was used for the *Arabidopsis* transformation[68]. The T1 generation transgenic plants were screened on the Hygromycin B selection medium. Positive plants were further confirmed by PCR and western blot using GFP antibodies (Cat. #SAB4301138, Sigma-Aldrich). The positive T2 generation transgenic plants were evaluated in a

growth chamber. Both an-HA antibody and GFP antibody were confirmed to be functional (Fig. S15).

**Plant materials used for investigation of the relationship between CLas infection, immune response, phloem blockage, cell death and HLB symptom development**. We used 2-year-old CLas-infected and healthy Valencia sweet orange (*Citrus sinensis*) plants maintained in a greenhouse (28 ± 2 °C, relative humidity of 50 ± 5%, natural light period). We selected young flushes from the twigs of HLB-positive sweet orange trees. Both "Valencia" and "Hamlin" sweet orange are *C. sinensis* and susceptible to HLB without observable differences in symptoms. The plant was infected with CLas by graft inoculation and maintained in a greenhouse. The sweet orange trees from the groves were naturally infected by CLas. Healthy plants were maintained in a glasshouse with natural light and without temperature control.

**Quantification of H$_2$O$_2$ concentrations**. H$_2$O$_2$ concentrations were quantified following the procedure described elsewhere[40]. CLas-positive asymptomatic mature leaves, mature leaves with mild or severe symptoms were collected from HLB-positive *C. sinensis* "Valencia" trees in citrus groves of Citrus Research and Education Center, University of Florida/Institute of Food and Agricultural Sciences. CLas negative mature leaves were collected from healthy *C. sinensis* "Valencia" trees in glasshouse. Briefly, leaf samples (0.5 g) were grinded in 0.1% (w/v) trichloroacetic acid (TCA) and centrifuged at 12,000 × *g* for 15 min at 4 °C. The supernatant (0.3 mL) was mixed with 1.7 mL 1.0 M potassium phosphate buffer (pH 7.0) and 1.0 mL of 1.0 M potassium iodide solution, then incubated for 5 min before measuring the absorbance of the oxidation product at 390 nm. H$_2$O$_2$ concentrations were calculated using a standard curve prepared with known concentrations of H$_2$O$_2$ and expressed in μmol/g fresh weight. For measuring H$_2$O$_2$ concentrations in the exudates of phloem-enriched bark tissues, the same procedure was used except the TCA step and H$_2$O$_2$ concentrations were expressed in mmol/L. CLas-positive symptomatic branches and CLas-negative branches from the spring flush were used for collection of bark tissues from the *C. sinensis* "Valencia" trees mentioned above.

**Ion leakage**. Conductivity of the exudates extracted from phloem-enriched bark tissues was measured using a CON 700 conductivity/°C/°F bench meter (Cat. #WD-35411-00, OAKTON Instruments, Vernon Hills, IL, USA).

**Callose deposition assay**. Leaf samples were fixed with FAA (37% formaldehyde/glacial acetic acid/95% ethanol/deionized water at a volume ratio of 50:5:10:35) solution overnight. Samples were embedded in the Tissue Plus O.C.T compound (Cat. #23-730-571, Thermo-Fisher, Waltham, MA, USA), sectioned with a Harris Cryostat Microtome (International Equipment, Boston, MA, USA) and stained with 0.005% aniline blue solution prior to analysis. Samples were observed in an Olympus BX61 epifluorescence microscope (Olympus Corporation, Center Valley, PA, USA). Callose spots were counted per slide area for all sample types.

**CLas quantification using qPCR**. Tissues (100 mg) were homogenized into powders using a TissueLyser II (Cat. #85300, Qiagen, Valencia, CA, USA). DNA was extracted using the DNeasy Plant kit (Cat. #69204, Qiagen), following the manufacturer's instructions, and eluted in 100 μL nuclease free water. DNA concentration was measured using a Synergy LX plate reader (BioTek, Winooski, VT, USA). Quantification of CLas in plant tissues was performed as described elsewhere[69]. Briefly, qPCR was carried out with primers and probe for CLas[70]. qPCR assays were performed with QuantiStudio3 (Cat. #A28567, Thermo-Fisher, Waltham, MA) using the Quantitec Probe PCR Master Mix (Cat. #204343, Qiagen) in a 25-μl reaction. The standard amplification protocol was 95 °C for 10 min followed by 40 cycles at 95 °C for 15 s and 60 °C for 60 s. All reactions were conducted in triplicate with CLas-positive and water controls. Quantification of CLas was conducted using the following equation:[71]

$$Y = -0.288 \times (CLas\ Ct) + 11.607$$

**Starch assay**. The samples (100 mg) were powdered using a TissueLyser II (Qiagen, Hilden, Germany). The powdered samples were used to quantify the starch. The starch estimation was performed using the Total Starch Assay Kit (AA/AMG) (Cat. #K-TSTA-100A, Megazyme, Bray, Ireland) as instructed by the manufacturer. The experiments were repeated thrice with similar result.

**Statistical analyses**. All statistical analyses were performed using SAS statistical software (Version 9.4, SAS Institute, Cary, NC, USA).

**Gene expression assays using reverse transcription quantitative PCR (RT-qPCR)**. Total RNA was extracted using the RNeasy Plant Mini Kit (Cat. #74904, Qiagen), according to manufacturer's instructions. cDNA was synthesized with Quantitect Reverse Transcription Kit (Cat. #205311, Qiagen) according to manufacturer's instructions and diluted 10 times for RT-qPCR. Reactions were carried

out by adding 1 μL of cDNA, 1 μL of each specific primer, 7 μL water and 10 μL Fast SYBR Green Master Mix (Cat. #4385617, Thermo-Fisher Scientific, Waltham, MA, USA) performed with QuantiStudio3 (Thermo-Fisher) using the standard fast protocol of 95 °C for 20 s followed by 40 cycles of 95 °C for 1 s and 60 °C for 20 s. Denaturation protocol consisted of 95 °C for 1 s, 60 °C for 20 s and a final dissociation step of 95 °C. Relative gene expression was calculated using the $2^{-\Delta\Delta C_T}$ method[72]. CsGAPDH was used as an endogenous control.

**TEM analysis**. Small sections of the leaf and stem samples were collected under a stereomicroscope (Swift Table Stereo Zoom Microscope, Carlsbad, CA, USA). The leaf samples were transferred to 3% glutaraldehyde overnight at 4 °C for fixation. Then, the samples were postfixed in 2% osmium tetroxide prepared in 3% glutaraldehyde for 4 h at room temperature in a fume hood. The samples were dehydrated by sequential treatment with 10, 20, 30, 40, 50, 60, 70, 80, 90, and 100% (thrice) acetone for 10 min each. The leaf samples were incubated sequentially in 50, 75, and 100% (twice) Spurr's low-viscosity epoxy resin prepared in acetone for 8 h each. One-micrometer sections were cut with glass knives using an ultra-microtome followed by staining with methylene blue/azure A for 30 s and basic fuchsin (0.1 g in 10 mL of 50% ethanol) for 30 s. The sections were observed under a Leitz Laborlux S compound microscope (Leica Microsystems, Wetzlar, Germany) for the right spot with a vascular system. The same blocks were trimmed with a surgical blade and then sectioned to 0.1 μm using a diamond knife under an ultramicrotome. The thin sections were collected on 200-mesh copper grids. The samples were stained with 2% aqueous uranyl acetate for 5 min, washed in water, and again stained with lead citrate followed by water wash. The micrographs were prepared and analyzed using a Morgagni 268 (FEI Company, Hillsboro, OR, USA) transmission electron microscope equipped with an AMT digital camera (Advanced Microscopy Techniques Corp., Danvers, MA, USA).

**Trypan-blue staining**. Trypan-blue staining was conducted as described by Fernández-Bautista et al.[73].

**Monitoring ROS formation and localization in phloem tissues by use of the fluorescent probe 2′,7′-dichlorodihydrofluorescein diacetate (H$_2$DCFDA) and confocal laser microscopy**. Four CLas-infected branches collected from HLB-positive *C. sinensis* "Valencia" trees and four CLas-free branches collected from healthy *C. sinensis* "Valencia" trees were collected and placed in glass test tube with 30 mL water containing 10 μM 2′,7′-dichlorodihydrofluorescein diacetate (H$_2$DCFDA) (Cat. #D399, Thermo-Fisher Scientific). Four leaves/branch at the top were kept to facilitate transpiration. Glass tubes were wrapped with aluminum foil and kept in room temperature for 24 h. Bark was peeled from the stem section that was submerged in water and placed on slide with inner side upwards. HLB-positive and healthy branches were also incubated in water without H$_2$DCFDA as controls. 2′,7′-dichlorofluorescein (DCF) fluorescence was visualized by confocal laser scanning microscopy (CLSM) (Leica TCS-SP5, Mannheim, Germany) with excitation/emission at 495 nm/525 nm.

**Trunk injection of HLB-positive 5-year-old *C. sinensis* trees**. Trunk injection was conducted as described elsewhere[37]. For each tree, approximately 0.4 g streptomycin sulfate (Cat.#AC612240500, laboratory grade; Thermo-Fisher Scientific) at 5 g/L was injected into the trunk of three trees. The amount of streptomycin injected was calculated to reach the concentration needed to kill CLas in planta based on the canopy volume[37].

**Exudates of phloem-enriched bark tissues for H$_2$O$_2$ and ion leakage assays**. Exudates of phloem-enriched bark tissues were extracted from stems following the procedure described elsewhere[74]. Stems were collected from small branches of spring spouts with mildly symptomatic leaves.

**Protoplast**. Protoplast cells of *C. sinensis* "Hamlin" were prepared as described by[75]. Embryogenic calli were subcultured on solid MT (Murashige and Tucker) media (Cat. #M5525, Phytotech) every 2 weeks. From the maintained calli suspension, cells were prepared and maintained in DOG liquid media as described elsewhere[76]. The final isolated protoplast cells were suspended in W5 solution (154 mM NaCl,125 mM CaCl2, 5 mM KCl, 2 mM MES at pH 5.7) at $1 \times 10^7$ cells/mL for different treatments.

**Treatment of protoplast cells with H$_2$O$_2$, antioxidants, and Gibberellic acid (GA)**. For H$_2$O$_2$ treatment, H$_2$O$_2$ was freshly prepared as 1 M stock solution with sterile double stilled water. For protoplast treatments with different concentrations of H$_2$O$_2$, H$_2$O$_2$ was further prepared as 100× stock with protoplast buffer (W5 solution). Protoplast cells were then treated with different concentrations of H$_2$O$_2$ for 24 h. For each treatment, at least three biological replicates were conducted. After 24 h treatment, samples were stained with Fluorescein Diacetate (FDA) (Cat. #F1303, Invitrogen) for viability observation. For 50 μL of sample, 2 μL of FDA was added. Immediately after staining, the samples were observed under Olympus BX53 Epi-fluorescence microscope with green channel. The ratio of green

cells (living cells) to total cells was calculated as viability rate. All steps and chemical treatments were performed at room temperature.

Protoplast cells co-treated with $H_2O_2$, antioxidants or Gibberellic acid (Alfa Aesar) were conducted as described above. Uric acid (Cat. # A13346, Thermo-Fisher Scientific) was dissolved in protoplast buffer (W5 solution). Rutin hydrate (Cat. # R5143, Sigma-Aldrich) was dissolved in DMSO as stock, Gibberellic acid (Cat. #A17843, Alfa Aesar) was dissolved in ethanol as stock.

**Treatment of citrus suspension culture cells with $H_2O_2$.** Sweet orang Hamlin suspension culture 7 days after subculture was used. Five mL of suspension culture cells were aliquoted into a 50-mL Falcon tube. Freshly prepared $H_2O_2$ was added into each tube at a concentration of 0 (water control), 0.6, 1.5, 1.8, or 3.6 mM. The tubes were incubated at room temperature with gentle shaking (100 rpm). Twenty-four hours after treatment, 50 μL of cells were pipetted into a 1.5-mL tube from each treatment. Each sample was stained with 2 μL of fluorescein diacetate (stain only living cells, green color) and 2 μL of propidium iodide (stain only dead cells, orange to red color). One minute after staining, the stained samples were observed under a fluorescent microscope with green and red dual channels.

**Foliar spray with antioxidants and GA.** Five-year-old Valencia sweet orange trees with similar symptoms were used for foliar spray treatments. All trees in the grove were HLB-positive. The experiment was a completely randomized design with 5 treatments. Each treatment consisted of four trees. The treatments were applied by foliar spray with 2.5 mL/L of Induce non-ionic surfactant (Helena Ag, Collier, TN, USA). One liter of solution per plant were applied at approximately 400 kPa using a handheld pump sprayer to apply on the whole tree. This pressure resulted a fine mist and was sufficient to produce runoff from the leaves to ensure complete coverage. The five treatments were applied to the various trees as follows: uric acid (1.8 mM), rutin (0.6 mM), GA (5 mg/L), GA (25 mg/L) with water as the negative control. Foliar spray was conducted in the evening to facilitate absorption. The chemicals GA, and uric acid were purchased from Fisher Scientific. Rutin was purchased from Sigma-Aldrich (St. Louis, MO, USA).

**GA treatment via foliar spray.** GA foliar spay was conducted in the first week of November, 2020. For the GA application, 20 ounces of Pro Gibb LV (Valent U.S.A. LLC, Walnut Creek, CA, USA) was mixed with water in a 100 Gal tank. 64 ounces of WIDESPREAD MAX (A.I. organosilicone) was included as the surfactant for leaf spray with airblast. Applications were conducted during night. One block of Valencia sweet orange on rootstock 942 was treated with GA, whereas the nearby Valencia/942 block was not treated with GA as a negative control. In addition, one block of Vernia sweet orange on X639 rootstock was treated with GA with one nearby Vernia/X639 block as a negative control. All blocks are 10 acres or more with approximately 140 trees/acre.

**RNA-seq analyses of GA treatment on citrus protoplast cells in the presence of $H_2O_2$.** Protoplast cells were prepared as described above and suspended in W5 solution (154 mM NaCl, 125 mM $CaCl_2$, 5 mM KCl, 2 mM MES at pH 5.7) at $1 \times 10^7$ cells/mL. All steps were performed at room temperature. The following two treatments were conducted: (1) Protoplast + $H_2O_2$ (1.8 μmol/mL) + Gibberellin (5 mg/L), and (2) Protoplast + $H_2O_2$ (1.8 μmol/mL). Protoplast cells were maintained at room temperature without shaking. RNA was collected at 6 h after treatment. Four biological replicates were included for each treatment.

Total RNA was extracted using the RNeasy plant kit (Qiagen, Valencia, CA), followed by treatment with RQ1 RNase-Free DNase (Cat. #M6101, Promega, Madison, WI). RNA concentration and quality were measured by a Nanodrop One Microvolume UV-Vis Spectrophotometer (Thermo-Fisher Scientific, Waltham, MA). Samples meeting the following requirements (Concentration ≥20 ng/μL, OD260/280 > 2.0) were sent to Novogene (Novogene, Davis CA) for cDNA libraries construction and RNA-seq analyses. Libraries were constructed with the NEBNext Ultra II RNA Library Prep Kit for Illumina (Illumina, San Diego, CA). Samples were sequenced to generate 150 bp paired-end reads using the Illumina NovaSeq 6000 platform (Illumina). Raw RNA-seq data were filtered by removing low-quality reads and adapters, and then aligned to the *Citrus sinensis* reference genome[77] using HISAT2 version 2.2.1[78] and SAMtools version 1.2[79] each gene was quantified using HTSeq-count version 0.11.2[80]. Different expressed genes (DEGs) analysis was performed using DESeq2 packages version 1.30.1 in R version 4.0.5[81]. Genes were considered significantly expressed with adjusted p value <0.05 (FDR method). The heatmap plots of expression profiling of DEGs were drawn using the pheatmap package version 1.0.12 in R program version 4.0.5[82].

**NADPH oxidase inhibitor diphenyleneiodonium (DPI) treatment on ROS levels in CLas-positive stems.** HLB-positive branches from the summer flush of Valencia sweet orange in the field were collected and then soaked in DPI (Cat. #AAJ64838MC, Thermo-Fisher) solution (25 μM) or water (control). After 48 h treatment, phloem-enriched bark tissues were collected for $H_2O_2$ concentration measurement. Experiments were repeated two times and representative result is shown.

**Evaluation of citrus tree growth and HLB symptoms in response to GA treatment.** Tree growth was evaluated by estimating trunk diameter, tree height (TH), and canopy volume (CV) on both GA treated and untreated control trees. For each treatment group, a total of 10 trees ($n = 10$) were randomly selected for the evaluation. A digital caliper (Fowler, Newton, MA) was used to take two measurements of trunk diameter (north-south and east-west orientation) at ~20 cm above the ground. A tape measure was used to measure the TH above the ground from the soil surface to the apical point of the plant. CV was estimated by taking the average of two independent measurements of the diameter of the canopy at different directions (north-south and east-west). The CV was estimated using the equation: $V = (2/3) \times p \times h \times (d/2)^2$, where h is the TH and d is the average diameter of the tree canopy[83]. All statistical analyses were performed using SAS V9.4 (SAS Institute Inc., Cary, NC). The data were first tested for normality and homogeneity of variance using the Shapiro–Wilk's test and Levene's test, respectively. A Student's two-tailed *t* test was performed to explore differences between GA treated and untreated control trees in growth performance traits.

HLB disease incidence in different treatments was evaluated by randomly checking 200 trees/treatment. Ratio of symptomatic leaves vs. total leaves in different treatments were investigated by evaluating three groups of branches/treatment with each group containing 16 branches that were selected randomly from 8 trees (2 branches/tree).

**Data analyses of previously published RNA-seq data between HLB-positive and healthy samples.** To generate the comprehensive expression pattern of sweet orange in response to CLas infection, we collected 15 microarray and 5 RNA-seq data sets from NCBI SRA and GEO databases (Table S2). The differentially expressed genes (DEGs) were determined using Limma version 3.46.0[84] and DESeq2 version 1.30.1[81] packages in R version 4.0.5 for microarray and RNA-seq data, respectively (adjusted p value <0.05 and |log2 fold change|>1). Gene ontology (GO) term enrichment of DEGs was conducted using agriGO v2.0: a GO analysis toolkit for the agricultural community[85] using the singular enrichment analysis tool. The heatmap plots were drawn using the gplots package version 3.1.1 in R program version 4.0.5[86].

**Reporting summary.** Further information on research design is available in the Nature Research Reporting Summary linked to this article.

## Data availability
The raw RNA-seq data generated in this study have been deposited in the NCBI Bioproject database under accession code PRJNA780217. Public transcriptomic data were obtained from NCBI GEO and SRA databases under accession numbers GSE33003, GSE33004, GSE33459, GSE29633, GSE33373, GSE101381, and SRP022979. Source data are provided with this paper.

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

## Acknowledgements

The research has been supported by Florida Citrus Initiative, Florida Citrus Research and Development Foundation, USDA National Institute of Food and Agriculture grant # 2018-70016-27412, #2016-70016-24833, and #2019-70016-29796. We thank Dr. Steven E. Lindow for critical reading of the paper.

## Author contributions

N.W. conceptualized, designed the experiments, and supervised the project. W.X.M. performed ROS related experiments; Z.Q.P. conducted ion leakage experiments; W.X.M. and Z.Q.P. conducted trypan-blue staining together; X.E.H. conducted research related to protoplast; S.S.P. tested H$_2$O$_2$ concentrations, callose deposition, and gene expression of young leaves; J.Y.L. conducted foliar spray and trunk injection; J.X. conducted bioinformatic analysis of transcriptomic data; D.A. conducted TEM analyses; F.V. conducted callose deposition analyses of mature leaves; Z.Q.P., F.V., and Y.X.H. conducted over-expression of CLas genes in citrus, tobacco and Arabidopsis; C.H. was involved in gene expression using RT-qPCR; W.T.W., Y.X.H., and J.Y.L. participated in ROS and ion leakage experiments; D.H.L. participated in field trials. D.S. conducted confocal microscopy analysis. N.W. wrote the paper with input from all co-authors.

## Competing interests

The authors declare no competing interests.
