## [Peer Review File · Nature Communications]

Reviewers' Comments:

Reviewer #1:

Remarks to the Author:

The manuscript by Ma et al. describes Citrus Huanglongbing as an immune-mediated disease where the effects of the disease are largely mediated by high accumulation of reactive oxygen species. The manuscript is overall very interesting and while some language editing is required well written. The work has potential to be a significant contribution to the field. However, some aspects of the work are confusing and some more experimental work is needed to explore the transcriptional misregulation of NADPH oxidases in the ROS accumulation associated with the disease.

I have some comments which could help to improve the work:

1. The authors aim to provide the view that Citrus Huanglongbing is an immune-mediated disease. I would strongly suggest to explain this much better to the reader. Much of the information is there but it is not easily extracted and understood. Are the authors proposing an "inflammation response" in the plant?
2. The authors focus on high accumulation of ROS in the phloem tissue. What is the origin of the ROS there? Later the authors describe strong transcriptional upregulation of NADPH oxidases (RBOH) and suggest that those enzymes would be the source of ROS. This should be tested as NADPH oxidase activity can be inhibited by application of the inhibitor DPI. This would be a relatively simple experiment. A more complicated experiment, which may not be realistic in citrus plants, would be silencing of the various NADPH oxidases. At this point, an inhibitor experiment would provide sufficient insight to support the involvement of NADPH oxidases.
3. Figure 5: The authors prepare protoplasts and treat them with hydrogen peroxide and observe cell death. I find this to be a problematic experiment as ROS at high amounts can be toxic but protoplasts, lacking their cell walls are a particularly fragile cell system. Could injection of ROS into the phloem provide a better experiment?

Minor comments:

1. Lines 74-77: The authors write that "Intriguingly, multiple characterized proteins of CLAs, such as SDE1, SDE15, and SahA, suppress plant immune response, suggesting that CLAs triggers immune response"... This is highly confusing, so does CLAs suppress the immune system or trigger it?
2. I would ask the authors to provide a brief overview over the lifestyle of the pathogen(e.g. is it a biotroph?) as far as currently known for the non-expert reader.
3. Lines 379-380: How do the authors defined ROS levels about a threshold to induce cell death? ROS have multifaceted roles not only in promoting cell death but also as signaling molecules.
4. I miss a reference to the work by Huang et al. 2021 (<https://www.pnas.org/content/118/6/e2019628118>). I wonder if the peptide would also reduce the high ROS accumulation in response to CLAs infection.

Reviewer #2:

Remarks to the Author:

1. Plant Innate immunity system was vaguely described in the introduction without substantial clarity between human and plant immunity systems.
2. Lines 53-55: No clarity on which one (CLas, Clam, and CLaf) is vectored by which insect.
3. Pitino et al. (2020) paper explains Clas alters H₂O₂ detoxification and upregulation of enzymes involved in the detoxification of radical ions is critical to increased HLB tolerance. Enhanced ROS leading to symptom development was also described.
Pitino, M., Armstrong, C. & Duan, Y. Molecular mechanisms behind the accumulation of ATP and H₂O₂ in citrus plants in response to 'Candidatus Liberibacter asiaticus' infection. *Hortic Res* 4, 17040 (2017). <https://doi.org/10.1038/hortres.2017.40>
4. Use of Gibberellic acid to treat HLB in citrus was reported by Lisa Tang and Tripti Vashisth. New insight in Huanglongbing-associated mature fruit drop in citrus and its link to oxidative stress, *Scientia Horticulturae*, Volume 265, 2020, 109246, ISSN 0304-4238, <https://doi.org/10.1016/j.scienta.2020.109246>.

5. I saw this abstract under the web link.

<https://www.biorxiv.org/content/10.1101/2021.08.15.456409v2>

6. Inadequate both literature review and references of existing knowledge on citrus HLB.

Overall, I reject this manuscript as in my opinion this manuscript does not significantly contribute to neither advancement of current knowledge on the citrus HLB disease nor to the disease management.

Reviewer #3:

Remarks to the Author:

This report by Ma et al. that HLB is an immune mediated disease potentially shines new light on this devastating disease that cost billions of dollars to the citrus industry annually. The study is quite comprehensive and logically compelling. Unfortunately, the supporting data, at least as presented, is not entirely convincing. Example are presented below.

1. Fig.1A-C: Starting day 15 there is a third column that is not define. I suspect it is CLas titer. Its it and if so state it.

2. Line 145: Where are the data supporting the statement that "CLas was observed in the phloem tissue"? Data are only shown for infected tissue, not specifically for phloem.

3. Fig.3 line 172: The authors state that "More cell death was observed with increasing CLas titer". However, in Fig.3 the CLas titer was HIGHER in asymptomatic tissue (29.4) than in mild (24.7) and severely (23.3) symptomatic tissue, while cell death was higher in mild and severely symptomatic tissue.

4a. Fig.4A-G: It is not possible to assess whether there is or is not more callose deposition is asymptomatic vs symptomatic tissue based on the data shown. The cumulative data from many pictures needs to be present in a quantitative manner.

4b. Fig.4H-J: The pictures are too dark to see cell structure and thus are uninterpretable.

4c. Fig.4K+L: CLas Ct is higher, rather than the expected lower. Thus, the experiment did not work or the data are mixed up.

4. In summary, the Fig.4 data, on which much of the conclusions are drawn, are not at all convincing due to lack of quantification, lack of visibility and contradictory numbers.

5. Line 279: Where are the data or reference supporting the claim that ROS production triggered by CLas is CHRONIC?

6. Line 304_314: The differential expression data are complex as correctly noted by the authors. However, overexpression of the NADPH oxidases is only evident is some of the data sets but not others. Hence, their claim is an overstatement that is not convincingly supported by the combined data. Thus, the statement in sentence starting on line 319 is not fully supported by the data as presented.

Other more minor problems/comments:

a. Fig. 5A and 6D: Rate should be replaced with percent of viable protoplast.

b. Fig.6: Why does GA suppress H₂O₂-induced protoplast death?

c. Line 89: What is the significance of increased starch accumulation?

Reviewer #4:

Remarks to the Author:

Abstract

The authors did not report on callus deposition in the abstract.

Please mention the genes that were up regulated due to HLB infections and if they responded with the mitigation treatments.

Check on sentence structure, ambiguity and try to apply scientific tone.

Line 16 is not necessary

Introduction

Sentence structure needs improvement.

The authors also need to improve on the background literature on immune-mediated diseases. A lot of emphasis has been put on human immune mediated diseases.

- What are the noteworthy results?

The manuscript highlights the importance of immune system in plants, against pathogens. The article stands out in being the first to report on immune-mediated disease for plants. The authors satisfactorily employed appropriate techniques to prove that citrus Huanglongbing (HLB), caused by phloem-colonizing *Candidatus Liberibacter asiaticus* (CLAs) is an immune-mediated disease. CLAs infection of *Citrus sinensis* stimulated systemic and chronic immune response in the phloem tissues including reactive oxygen species (ROS) production as indicated by H₂O₂, callose deposition, and induction of immune related genes. Systemic cell death of companion and sieve element cells, but not surrounding parenchyma cells, was observed following ROS production triggered by CLAs. ROS production triggered by CLAs localized in phloem tissues. The H₂O₂ concentration in exudates extracted from RNA-seq analyses of CLAs infected and health *C. sinensis* support that CLAs causes oxidative stress. In sum, HLB is an immune-mediated disease and both mitigating ROS via antioxidants and promoting plant growth can reduce cell death of the phloem tissues caused by CLAs, thus controlling HLB.

- Will the work be of significance to the field and related fields? How does it compare to the established literature? If the work is not original, please provide relevant references.

This work provides fundamental information underlying immune-mediated diseases in plants. In established literature this work provides an integral part of plant pathology as it is the first reported in literature. This therefore underpins the originality of the study area as well as its significant contribution in plant immunology.

- Does the work support the conclusions and claims, or is additional evidence needed?

The tools, experimental methodologies and techniques employed and literature reviewed by the authors sufficiently support the conclusions and the claims. However, the authors' background literature was mainly drawn from human studies.

- Are there any flaws in the data analysis, interpretation and conclusions? Do these prohibit publication or require revision?

The data analysis was properly done. The interpretation of the subsequent results was okay and their interpretation was backed by previous findings. However the authors should provide adequate background information on the reported symptoms of *Candidatus Liberibacter asiaticus* (CLAs) for a grounded comparison of their research observations and findings.

- Is the methodology sound? Does the work meet the expected standards in your field?

The general methodology is acceptable, but for the experimental design and treatment application on plant materials used for investigation of the relationship between CLAs infection, immune response, phloem blockage, cell death and HLB symptom development is not clear. The authors need to clearly elaborate. How many plants/shoots were applied?

For quantification of H₂O₂ concentrations, the leaf samples were collected from which part of the plant (i.e young recently emerged leaf or old leaves?)

Generally the authors' work is of standard and is publishable.

- Is there enough detail provided in the methods for the work to be reproduced?

The methods used by the authors are scientifically sound but they should elaborate their methods to be reproducible.

It is also notable that the Rna-seq data had been done earlier. Was it published? If so, the authors need to provide the repository number.

Additionally: The paper needs to be reviewed for grammar and sentence structure.

RESPONSE TO REVIEWERS' COMMENTS

Reviewer #1 (Remarks to the Author):

The manuscript by Ma et al. describes Citrus Huanglongbing as an immune-mediated disease where the effects of the disease are largely mediated by high accumulation of reactive oxygen species. The manuscript is overall very interesting and while some language editing is required well written. The work has potential to be a significant contribution to the field. However, some aspects of the work are confusing and some more experimental work is needed to explore the transcriptional misregulation of NADPH oxidases in the ROS accumulation associated with the disease.

Response: We thank the reviewer for the encouraging comments. It is always inspiring to know colleagues appreciate our work. We also appreciate the critical and constructive comments and have conducted additional experiments and revised the manuscript as suggested.

1. The authors aim to provide the view that Citrus Huanglongbing is an immune-mediated disease. I would strongly suggest to explain this much better to the reader. Much of the information is there but it is not easily extracted and understood. Are the authors proposing an "inflammation response" in the plant?

Response: We have significantly revised the manuscript to provide more background information.

Our data suggest that there are some similarities between CLas induced damage to citrus and inflammation diseases of human, especially sepsis.

2. The authors focus on high accumulation of ROS in the phloem tissue. What is the origin of the ROS there? Later the authors describe strong transcriptional upregulation of NADPH oxidases (RBOH) and suggest that those enzymes would be the source of ROS. This should be tested as NADPH oxidase activity can be inhibited by application of the inhibitor DPI. This would be a relatively simple experiment. A more complicated experiment, which may not be realistic in citrus plants, would be silencing of the various NADPH oxidases. At this point, an inhibitor experiment would provide sufficient insight to support the involvement of NADPH oxidases.

Response: Excellent suggestions. We are generating RbohB, D, and F mutants via gene editing. Meanwhile, we are silencing RbohB, D, and F. For citrus, it takes approximately three years to get the mutants and conduct CLas inoculation.

We tested DPI inhibition of ROS production via injection into the leaf and keeping branches in the DPI solution. As shown in Fig. S10, DPI significantly inhibits ROS production. We thank the reviewer for this suggestion.

3. Figure 5: The authors prepare protoplasts and treat them with hydrogen peroxide and observe cell death. I find this to be a problematic experiment as ROS at high amounts can be toxic but protoplasts, lacking their cell walls are a particularly fragile cell system. Could injection of ROS into the phloem provide a better experiment?

Response: Because the H₂O₂ concentrations were determined for phloem sap, protoplast cells without cell wall allow H₂O₂ to enter the cytoplasm easier than cells with cell wall, thus better mimicking the direct effect of ROS on plant cells. In addition, addition of uric acid (0.2 mM), a ROS scavenger, reduced both cell death caused by H₂O₂ (Figs. 5, A and B), indicating the cell death of protoplast cells does not result from the "fragile nature".

Anyway, to address the concern regarding "fragile nature" of protoplast, we repeated the experiment with suspension culture as shown in Fig. S7. Suspension culture experiment showed similar results as the protoplast experiment.

Phloem is notoriously challenging to manipulate. We could not find a method to directly inject ROS into the phloem.

Minor comments:

1. Lines 74-77: The authors write that "Intriguingly, multiple characterized proteins of CLAs, such as SDE1, SDE15, and SahA, suppress plant immune Response, suggesting that CLAs triggers immune response"... This is highly confusing, so does CLAs suppress the immune system or trigger it?

Response: The current data suggest that PAMPs of CLAs trigger excessive immune response including ROS production, which is also detrimental to CLAs. CLAs seems to have some mechanism to suppress immune response for its own benefit. However, it seems such a suppression mechanism is not enough to suppress the excessive immune response. With that being said, we agree with the concerns and have removed them and revised the sentence.

2. I would ask the authors to provide a brief overview over the lifestyle of the pathogen(e.g. is it a biotroph?) as far as currently known for the non-expert reader.

Response: A brief overview over the lifestyle of the pathogen was added in the 2nd paragraph of the introduction as follows (Line 59-68): Citrus Huanglongbing (HLB, also known as citrus greening) is currently the most devastating citrus disease, causing billions of dollars of economic losses worldwide annually ^{10,11}. HLB presents an unprecedented challenge for the citrus industry despite some promising progress in research ¹²⁻¹⁴. HLB is caused by the phloem-colonizing *Candidatus Liberibacter asiaticus* (CLAs), *Ca. L. americanus* and *Ca. L. africanus* with CLAs being the most prevalent in the world ¹⁵. CLAs is a biotroph, vectored by the Asian citrus psyllid (*Diaphorina citri*), and in the Rhizobiaceae family. Its approximately 1.23 Mb genome is significantly reduced compared to other free-living members of the Rhizobiaceae family, resulting from the reductive evolution within the nutrient-rich phloem of citrus and psyllid tissues ¹⁶.

3. Lines 379-380: How do the authors defined ROS levels about a threshold to induce cell death? ROS have multifaceted roles not only in promoting cell death but also as signaling molecules.

Response: At high concentrations, ROS triggers necrotic cell death, but induces programmed cell death below the ROS threshold. It seems ROS induction correlates with CLAs titers in the local tissues. Due to the challenges to measure other ROS, we use H₂O₂ as an indicator. As shown in Fig. 5, the commonly observed H₂O₂ concentration at 1.8 μmol/mL in the phloem sap is sufficient to kill protoplast cells, which is inhibited by the ROS scavenger uric acid, indicating ROS might be able to kill citrus phloem cells. On the hand, your suggestion is insightful because ROS are signaling molecules that trigger programmed cell death, which might also contribute to the cell death of phloem tissue. ROS also inhibit plant growth. We added this

point in the discussion (line 433-437): ROS concentrations triggered by CLas infection are above the threshold needed to induce cell death, probably resulting from the combined effect of programmed cell death induced at low ROS concentrations and necrotic cell death stimulated at high ROS concentrations. In addition, ROS positively regulate callose deposition^{24,52} and inhibit plant growth including roots^{53,54}.

4. I miss a reference to the work by Huang et al. 2021 (<https://www.pnas.org/content/118/6/e2019628118>). I wonder if the peptide would also reduce the high ROS accumulation in response to CLas infection.

Response: Thanks for the suggestion. We have added the reference in the introduction. Line 61-62: HLB presents an unprecedented challenge for the citrus industry despite some promising progress in research¹²⁻¹⁴.

Reviewer #2:

1. Plant Innate immunity system was vaguely described in the introduction without substantial clarity between human and plant immunity systems.

Response: More information were provided as suggested as follows (Line 40-52): Both plants and animals have innate immunity, whereas animals also have adapted immunity. The plant innate immune system consists of pattern-triggered immunity (PTI), which is triggered by pathogen-associated molecular patterns (PAMPs) via cell surface-localized pattern-recognition receptors (PRRs), and effector-triggered immunity (ETI), which is instigated by pathogen effector proteins via intracellular receptors called nucleotide-binding, leucine-rich repeat receptors (NLRs)¹⁻³. However, some human diseases are mediated by the immune response itself, including autoimmune diseases such as inflammatory bowel disease, and non-autoimmune diseases, like asthma, sepsis induced by various microbes⁴, and cryptococcal meningitis caused by *Cryptococcus neoformans*⁵. Host immune responses have been known to be an important factor in addition to microbial pathogenicity factors for human diseases caused by microbial pathogens, which has resulted in the proposal of a damage-response framework of microbial pathogenesis⁶. Immune-mediated diseases have not been reported in the Plantae Kingdom.

2. Lines 53-55: No clarity on which one (CLas, CLam, and CLaf) is vectored by which insect.

Response:

We have revised the sentence as follows (line 62-65): HLB is caused by the phloem-colonizing *Candidatus Liberibacter asiaticus* (CLas), *Ca. L. americanus* and *Ca. L. africanus* with CLas being the most prevalent in the world¹⁵. CLas is a biotroph, vectored by the Asian citrus psyllid (*Diaphorina citri*), and in the Rhizobiaceae family.

3. Pitino et al. (2020) paper explains CLas alters H₂O₂ detoxification and upregulation of enzymes involved in the detoxification of radical ions is critical to increased HLB tolerance. Enhanced ROS leading to symptom development was also described.

Pitino, M., Armstrong, C. & Duan, Y. Molecular mechanisms behind the accumulation of ATP and H₂O₂ in citrus plants in response to 'Candidatus Liberibacter asiaticus' infection. *Hortic Res* 4, 17040 (2017). <https://doi.org/10.1038/hortres.2017.40>

Response: We think the reviewer referred to Pitino et al. 2017.

In the study by Pitino et al. 2017, the authors investigated H₂O₂ and ATP accumulation in relation to citrus Huanglongbing (HLB) in addition to revealing the expression profiles of genes critical for the production and detoxification of H₂O₂ and ATP synthesis. They found that as ATP and H₂O₂ concentrations increased in the leaf, so did the severity of the HLB symptoms, a trend that remained consistent among the four different citrus varieties tested. Furthermore, the upregulation of ATP synthase, a key enzyme for energy conversion, may contribute to the accumulation of ATP in infected tissues, whereas downregulation of the H₂O₂ detoxification system may cause oxidative damage to plant macromolecules and cell structures. This may explain the cause of some of the HLB symptoms such as chlorosis or leaf discoloration.

This study by Pitino et al. 2017 had some interesting findings between the association of CLAs infection, H₂O₂ concentrations, and HLB disease severity. We have cited this study as follows (line 227-228): We observed significantly higher H₂O₂ concentrations in CLAs-infected mature leaves than CLAs-free leaves (Fig. 3B), consistent with a previous study by Pitino et al. ²².

Our study is distinct from Pitino et al. 2017 because of the following: In our study, we present evidence that citrus Huanglongbing (HLB) is an immune-mediated disease. CLAs infection of *Citrus sinensis* stimulated systemic and chronic immune response in the phloem tissues including reactive oxygen species (ROS) production as indicated by H₂O₂, callose deposition, and induction of immune related genes. Systemic cell death of companion and sieve element cells, but not surrounding parenchyma cells, was observed following ROS production triggered by CLAs. ROS production triggered by CLAs localized in phloem tissues. The H₂O₂ concentration in exudates extracted from phloem enriched bark tissue of CLAs infected plants reached a threshold of killing citrus protoplast cells, which was suppressed by uric acid (a ROS scavenger) and gibberellin. Foliar spray of HLB positive citrus with antioxidants (uric acid and rutin) significantly reduced both H₂O₂ concentrations and cell death in phloem tissues induced by CLAs and reduced HLB symptoms. RNA-seq analyses of CLAs infected and health *C. sinensis* support that CLAs causes oxidative stress. In sum, HLB is an immune-mediated disease and both mitigating ROS via antioxidants and promoting plant growth can reduce cell death of the phloem tissues caused by CLAs, thus controlling HLB.

4. Use of Gibberellic acid to treat HLB in citrus was reported by Lisa Tang and Tripti Vashisth. New insight in Huanglongbing-associated mature fruit drop in citrus and its link to oxidative stress, *Scientia Horticulturae*, Volume 265, 2020, 109246, ISSN 0304-4238, <https://doi.org/10.1016/j.scienta.2020.109246>.

Response: No, Tang and Vashisth 2020 did not report using Gibberellic acid to treat HLB in citrus. Our study is distinct from Tang and Vashisth 2020 that focused on fruit drop caused by HLB. Tang and Vashisth 2020 has been cited as follows (line 352-353): Intriguingly, the involvement of oxidative stress in HLB disease has been suggested by multiple previous studies ^{22,42,43}.

Below please find the abstract of Tang and Vashisth 2020 as mentioned by the reviewer.

As Huanglongbing (HLB) becomes epidemic throughout Florida, there is a sharp rise in mature fruit drop prior to harvest in affected citrus, leading to a substantial reduction in yield. To investigate how HLB increases mature fruit abscission, this research evaluated the drop rate and compared the global gene expression using RNA sequencing in HLB-susceptible 'Hamlin' sweet orange (*Citrus sinensis*) versus HLB-tolerant 'LB8-9' mandarin { 'Clementine' mandarin (*C. reticulata*) and 'Minneola' tangelo [hybrid of 'Duncan' grapefruit (*C. paradisi*) and 'Dancy' tangerine (*C. reticulata*)]}. Consistent with their susceptibility to HLB documented in the literature, the drop rate of mature fruit was significantly higher in 'Hamlin' sweet orange than

'LB8-9' mandarin. Between the two citrus, there were 368 differentially expressed genes (DEGs); 310 and 58 were upregulated and downregulated, respectively, in the abscission zone (AZ) of 'Hamlin' sweet orange compared to 'LB8-9' mandarin fruit. Although at the time of collection, fruits of both citrus were still attached to the tree branch (fruit detachment force > 6 kgf), upregulation of the DEGs related to cell wall loosening in 'Hamlin' sweet orange indicates that the process of abscission was already initiated in the AZ of this citrus. The pattern of DEGs related to the metabolism and signal transduction of hormones, specifically ethylene, auxin, cytokinin, abscisic acid, and gibberellic acid, also suggests that HLB likely brings about changes in endogenous hormone balance that promote mature fruit abscission. Interestingly, DEGs encoding antioxidants were upregulated, suggesting a demand for detoxification mechanisms against oxidative stress in the AZ of 'Hamlin' sweet orange fruit. Additionally, gene ontology (GO) terms for cell death and senescence, two of consequences of oxidative stress, were highly enriched. Together, the results of transcriptome analysis presented herein provide evidence that for HLB-susceptible 'Hamlin' sweet orange, oxidative stress caused by the pathogen infection likely results in cell wall modification, leading to cell separation and eventually fruit abscission. The results further suggest that 'LB8-9' mandarin may have an advanced antioxidant system to mitigate the pathogen-induced oxidative stress, thereby contributing to its tolerance to HLB.

5. I saw this abstract under the web link.

<https://www.biorxiv.org/content/10.1101/2021.08.15.456409v2>

Response: Nature journals support preprints. We have posted our manuscript as a preprint in biorxiv to facilitate the spread of knowledge.

6. Inadequate both literature review and references of existing knowledge on citrus HLB.

Response: We have provided more literature review and references of existing knowledge on citrus HLB as suggested.

Overall, I reject this manuscript as in my opinion this manuscript does not significantly contribute to neither advancement of current knowledge on the citrus HLB disease nor to the disease management.

Response: We respectively disagree with the assessment that "not significantly contribute to neither advancement of current knowledge on the citrus HLB disease nor to the disease management" because of the following reasons:

First, as you know, immune-mediated diseases are common for human, but have not been previously known for plants. This study provides the evidence that CLAs infection of the phloem tissues of citrus triggers immune response including ROS, callose deposition and immune genes. H₂O₂ concentration triggered by CLAs reaches the threshold to directly kill citrus cells. Cell death due to ROS triggered by CLAs was suppressed by treatment with antioxidants (uric acid and rutin). We have established the causative relationship that CLAs triggers ROS production in the phloem tissues, which subsequently causes cell death of phloem tissues, leading to HLB disease symptoms. This study established that immune-mediated diseases also happen in the Plantae Kingdom.

Second, citrus HLB is the most detrimental disease worldwide causing billions of dollars in annual economic losses and has devastated the citrus industry in Florida. We have shown that mitigating cell death caused by CLAs via antioxidants or immunoregulator gibberellin and promoting plant growth with gibberellin halts or reduces HLB symptoms. This study delivers an

immediate solution to the notorious HLB disease, but also provides strategic guidance regarding how to develop long-term solution to control HLB through genetic improvements.

Finally, the finding that HLB being an immune mediated disease explains most known HLB phenomena. For instance, phloem dysfunction resulting from death of companion and sieve element cells may lead to starch accumulation, and blotchy mottle symptoms. Hardened leaves might result from the action of ROS since ROS are known to directly cause strengthening of host cell walls. Both cell death of the phloem tissues and reduced transportation of photosynthates may be responsible for root decay. Stunt growth probably results from the direct effect of ROS, reduced transportation of carbohydrates and hormones. In addition, it was reported that HLB-tolerant varieties contain higher levels of antioxidant capacities than the susceptible varieties.

Overall, this new work represents a major advance toward understanding the mysterious nature of how CLas causes HLB disease, which my lab has been working on for the last 14 years (<https://scholar.google.com/citations?user=Owf0xG8AAAAJ&hl=en>). This work also represents a breakthrough in immune-mediated plant disease.

Reviewer #3 (Remarks to the Author):

This report by Ma et al. that HLB is an immune mediated disease potentially shines new light on this devastating disease that cost billions of dollars to the citrus industry annually. The study is quite comprehensive and logically compelling. Unfortunately, the supporting data, at least as presented, is not entirely convincing. Example are presented below.

1. Fig.1A-C: Starting day 15 there is a third column that is not define. I suspect it is CLas titer. Its it and if so state it.

Response: Revised as suggested.

2. Line 145: Where are the data supporting the statement that "CLas was observed in the phloem tissue"? Data are only shown for infected tissue, not specifically for phloem.

Response: As shown in Fig. 2C, D, and E, the phloem tissue is indicated by the arrows. Transmission Electron Microscopy (TEM) is a technology that can clearly differentiate phloem tissue from other tissue types.

3. Fig.3 line 172: The authors state that "More cell death was observed with increasing CLas titer". However, in Fig.3 the CLas titer was HIGHER in asymptomatic tissue (29.4) than in mild (24.7) and severely (23.3) symptomatic tissue, while cell death was higher in mild and severely symptomatic tissue.

Response: For qPCR, lower Ct values indicate higher bacterial titers. We have added one sentence in the figure legend to help readers with this: "CLas titers were determined by qPCR as shown by Ct values with lower Ct values indicating higher bacterial titers." in Fig. 3.

4a.Fig.4A-G: It is not possible to assess whether there is or is not more callose deposition is asymptomatic vs symptomatic tissue based on the data shown. The cumulative data from many pictures needs to be present in a quantitative manner.

Response: Fig. S5 (CLas infection induces callose deposition in phloem tissues of *C. sinensis*) was generated in a quantitative manner as suggested.

4b. Fig.4H-J: The pictures are too dark to see cell structure and thus are uninterpretable.

Response: We have adjusted the contrast of the pictures. In addition, the tissues used are bark tissues which are phloem enriched. We have changed the description to be in "phloem-enriched bark tissues".

4c. Fig4K+L: CLas Ct is higher, rather than the expected lower. Thus, the experiment did not work or the data are mixed up.

Response: For qPCR, lower Ct values indicate higher bacterial titers. We have added one sentence in the figure legend in Fig. 3 to help readers with this: "CLas titers were determined by qPCR as shown by Ct values with lower Ct values indicating higher bacterial titers". I think it is better not to state here again in Fig. 4.

4. In summary, the Fig.4 data, on which much of the conclusions are drawn, are not at all convincing due to lack of quantification, lack of visibility and contradictory numbers.

Response: We have responded to the raised issues above.

5. Line 279: Where are the data or reference supporting the claim that ROS production triggered by CLas is CHRONIC?

Response: As shown in Fig. 1A, CLas infection triggers ROS production (as indicated by H₂O₂) as early as 15 days post bud initiation. As shown in Fig. 3B, ROS production as indicated by H₂O₂ was observed in both asymptomatic, and leaves with mild or severe symptoms resulting from CLas infection. In addition, CLas induction of ROS is consistently observed in all the samplings over the year.

To address the reviewer's concern, we have revised the sentences as follows (line 291-296): Instead, high levels of ROS were consistently detected in both young leaves during early stages of infection (Fig. 1A) as well as in mature infected leaves and stems (Fig. 3B-C), probably triggered by CLas colonizing and multiplying in the previously unoccupied phloem tissue as well as that triggered by damage-associated molecular patterns resulting from dying companion and sieve element cells.

6. Line 304_314: The differential expression data are complex as correctly noted by the authors. However, overexpression of the NADPH oxidases is only evident in some of the data sets but not others. Hence, their claim is an overstatement that is not convincingly supported by the combined data. Thus, the statement in sentence starting on line 319 is not fully supported by the data as presented.

Response: We have revised the statement as follows (line 323-328): Critically, expression of respiratory burst oxidative homolog D (RBOHD) gene, which encode NADPH oxidase implicated in the generation of ROS during defense responses, was induced by CLas infection in several studies. RBOHD is primarily responsible for ROS produced upon PAMP recognition and is required for cell death that is initiated after pathogen detection⁴⁰. In addition, RBOHB and RBOHF have also been reported to be involved in ROS production in response to pathogen infection^{40,41}.

To further address the concerns, we have conducted qRT-PCR assays for RBOHB, D, and

F in healthy and HLB positive leaves. The data are shown in line 328-331: qRT-PCR assays revealed that both *RBOHB* and *RBOHD* were induced by CLas infection of leaf samples collected in both field and greenhouse conditions, whereas *RBOHF* was induced only under field conditions (fig. S9).

Other more minor problems/comments:

a. Fig. 5A and 6D: Rate should be replaced with percent of viable protoplast.

Response: Revised. I think Reviewer might refer 5B as 5A here.

b. Fig.6: Why does GA suppress H₂O₂-induced protoplast death?

Response: Excellent question.

To address this question, we have conducted a RNA-seq analysis of GA and non-GA treated *Citrus sinensis* protoplast cells in the presence of 1.8 mM H₂O₂. As shown in line 369-374: RNA-seq analyses of GA treated vs. non-GA treated *C. sinensis* protoplast cells in the presence of 1.8 mM H₂O₂ demonstrated that GA induced the expression of genes encoding H₂O₂ scavenging enzymes catalases, ascorbate peroxidases, and glutathione peroxidases. GA also inhibited the expression of *RbohD* (Fig. 7, Table S4). GA treatment therefore clearly alleviates the oxidative stress caused by H₂O₂ in citrus.

c. Line 89: What is the significance of increased starch accumulation?

Response: Starch accumulation was suggested to result in disruption of chloroplast inner grana structure, contributing to the yellowing symptoms. Our current model is that CLas triggers cell death of phloem tissues, which blocks phloem transportation. Consequently, starch accumulation happens that contributes to the HLB symptoms.

Reviewer #4:

Abstract

The authors did not report on callus deposition in the abstract.

Response: In line 20-21, we did mention: ...**as indicated by H₂O₂, callose deposition...**

Please mention the genes that were up regulated due to HLB infections and if they responded with the mitigation treatments.

Response: We have added the following (line 32-36): RNA-seq analyses of CLas infected and healthy *C. sinensis* demonstrates that CLas infections induce the expression of genes encoding NADPH oxidases and triggers downregulation of antioxidant enzyme genes, supporting that CLas causes oxidative stress. CLas also stimulate the expression of immune related genes.

Check on sentence structure, ambiguity and try to apply scientific tone.

Response: We have revised the manuscript carefully and checked by two native English speakers including Dr. Steven E. Lindow at UC Berkeley and Dr. Connor Hendrich from my lab.

Line 16 is not necessary

Response: Removed.

Introduction

Sentence structure needs improvement.

The authors also need to improve on the background literature on immune-mediated diseases. A lot of emphasis has been put on human immune mediated diseases.

Response: We have revised the section as suggested.

- What are the noteworthy results?

The manuscript highlights the importance of immune system in plants, against pathogens. The article stands out in being the first to report on immune-mediated disease for plants. The authors satisfactorily employed appropriate techniques to prove that citrus Huanglongbing (HLB), caused by phloem-colonizing *Candidatus Liberibacter asiaticus* (CLAs) is an immune-mediated disease.

CLAs infection of *Citrus sinensis* stimulated systemic and chronic immune response in the phloem tissues including reactive oxygen species (ROS) production as indicated by H₂O₂, callose deposition, and induction of immune related genes. Systemic cell death of companion and sieve element cells, but not surrounding parenchyma cells, was observed following ROS production triggered by CLAs. ROS production triggered by CLAs localized in phloem tissues. The H₂O₂ concentration in exudates extracted from RNA-seq analyses of CLAs infected and health *C. sinensis* support that CLAs causes oxidative stress. In sum, HLB is an immune-mediated disease and both mitigating ROS via antioxidants and promoting plant growth can reduce cell death of the phloem tissues caused by CLAs, thus controlling HLB.

Response: We really appreciate the positive comments. It is encouraging to know our colleague appreciates our work.

- Will the work be of significance to the field and related fields? How does it compare to the established literature? If the work is not original, please provide relevant references.

This work provides fundamental information underlying immune-mediated diseases in plants. In established literature this work provides an integral part of plant pathology as it is the first reported in literature. This therefore underpins the originality of the study area as well as its significant contribution in plant immunology.

Response: We thank the reviewer again for the encouraging comments.

- Does the work support the conclusions and claims, or is additional evidence needed?

The tools, experimental methodologies and techniques employed and literature reviewed by the authors sufficiently support the conclusions and the claims. However, the authors' background literature was mainly drawn from human studies.

Response: We have added more information regarding plant immunity as follows (line 42-52): Both plants and animals have innate immunity, whereas animals also have adapted immunity. The plant innate immune system consists of pattern-triggered immunity (PTI), which is triggered by pathogen-associated molecular patterns (PAMPs) via cell surface-localized pattern-recognition receptors (PRRs), and effector-triggered immunity (ETI), which is instigated by pathogen effector proteins via intracellular receptors called nucleotide-binding, leucine-rich repeat receptors (NLRs)¹⁻³. However, some human diseases are mediated by the immune response itself, including autoimmune diseases such as inflammatory bowel disease, and non-autoimmune diseases, like asthma, sepsis induced by various microbes⁴, and cryptococcal meningitis caused by *Cryptococcus neoformans*⁵. Host immune responses have been known to be an important factor in addition to microbial pathogenicity factors for human diseases caused

by microbial pathogens, which has resulted in the proposal of a damage-response framework of microbial pathogenesis⁶. Immune-mediated diseases have not been reported in the Plantae Kingdom.

- Are there any flaws in the data analysis, interpretation and conclusions? Do these prohibit publication or require revision?

The data analysis was properly done. The interpretation of the subsequent results was okay and their interpretation was backed by previous findings. However the authors should provide adequate background information on the reported symptoms of *Candidatus Liberibacter asiaticus* (CLas) for a grounded comparison of their research observations and findings.

Response: In the introduction, we further revised the HLB symptoms with more details as follows (line 70-74): No pathogenicity factors have been confirmed to be responsible for the HLB symptoms of characteristic blotchy mottle on leaves, hardened leaves, small and upright leaves, leaves showing zinc or manganese deficiency, corky veins, twig dieback, stunted growth of seedlings, thin canopy, small and lopsided fruit and root decay¹⁵.

In the result section, we have made the following changes (highlighted in the text):

Line 375-378: Six weeks after initiation of foliar sprays, treated plants exhibited reduced HLB symptoms (i.e., less blotchy mottle) compared with that of plants before treatment, whereas the blotchy mottling symptoms of plants treated only with water became more severe in this period (fig. S12).

Line 411-419: GA treatment significantly reduced the incidence of symptomatic leaves (i.e., leaves showing blotchy mottle, yellowing, and nutrient deficiency) (Fig. 6, E and G, and fig. S13), suggesting reduced death of sieve element and companion cells in treated leaves. In addition, foliar sprays of GA on HLB-positive 6-year-old *C. sinensis* var. 'Valencia' and var. 'Vernia' significantly promoted plant growth as measured by tree height, trunk diameter and canopy volume when assessed eight months after application (fig. S14). We presume GA reduces HLB symptoms via its direct effect on both mitigating ROS (Figs. 6A, C & D and 7) and promoting plant growth (fig. S14), thus alleviating the growth inhibition caused by CLas.

- Is the methodology sound? Does the work meet the expected standards in your field?

The general methodology is acceptable, but for the experimental design and treatment application on plant materials used for investigation of the relationship between CLas infection, immune response, phloem blockage, cell death and HLB symptom development is not clear. The authors need to clearly elaborate. How many plants/shoots were applied?

Response: We have provided the information and highlighted the details of treatments for foliar spray.

Line 661-681:

Foliar spray with antioxidants and GA

Five-year-old Valencia sweet orange trees with similar symptoms were used for foliar spray treatments. All trees in the grove were HLB-positive. The experiment was a completely randomized design with 5 treatments. Each treatment consisted of four trees. The treatments were applied by foliar spray with 2.5 mL/L of Induce non-ionic surfactant (Helena Ag, Collier, TN, USA). One liter of solution per plant were applied at approximately 400 kPa using a handheld pump sprayer to apply on the whole tree. This pressure resulted a fine mist and was sufficient to produce runoff from the leaves to ensure complete coverage. The five treatments were applied to the various trees as follows: uric acid (1.8 mM), rutin (0.6 mM), GA (5 mg/L),

GA (25 mg/L) with water as the negative control. Foliar spray was conducted in the evening to facilitate absorption. The chemicals GA, and uric acid were purchased from Fisher Scientific. Rutin was purchased from Sigma-Aldrich (St. Louis, MO, USA).

GA treatment via foliar spray

GA foliar spray was conducted in the first week of November, 2020. For the GA application, 20 ounces of Pro Gibb LV (Valent U.S.A. LLC, Walnut Creek, CA, USA) was mixed with water in a 100 Gal tank. 64 ounces of WIDESPREAD MAX (A.I. organosilicone) was included as the surfactant for leaf spray with airblast. Applications were conducted during night. One block of Valencia sweet orange on rootstock 942 was treated with GA, whereas the nearby Valencia/942 block was not treated with GA as a negative control. In addition, one block of Vernia sweet orange on X639 rootstock was treated with GA with one nearby Vernia/X639 block as a negative control. **All blocks are 10 acres or more with approximately 140 trees/acre.**

For quantification of H₂O₂ concentrations, the leaf samples were collected from which part of the plant (i.e young recently emerged leaf or old leaves?)

Response: The following information was added and highlighted in the revision (line 531-534): CLas positive asymptomatic mature leaves, mature leaves with mild or severe symptoms were collected from HLB-positive *C. sinensis* 'Valencia' trees in citrus groves of Citrus Research and Education Center, University of Florida/Institute of Food and Agricultural Sciences. CLas negative mature leaves were collected from healthy *C. sinensis* 'Valencia' trees in glasshouse. Line 543-545: CLas positive symptomatic branches and CLas-negative branches from the spring flush were used for collection of bark tissues from the *C. sinensis* 'Valencia' trees mentioned above.

Generally the authors' work is of standard and is publishable.

Response: Thanks.

- Is there enough detail provided in the methods for the work to be reproduced?

The methods used by the authors are scientifically sound but they should elaborate their methods to be reproducible.

It is also notable that the Rna-seq data had been done earlier. Was it published? If so, the authors need to provide the repository number.

Response: Yes, the detailed information including repository number is shown in Table S2: Transcriptomic studies of sweet orange in response to CLas infection that were used for GO enrichment analysis in this study.

Additionally: The paper needs to be reviewed for grammar and sentence structure.

Response: We have revised the manuscript extensively and it has been checked by two native English speakers.

Reviewers' Comments:

Reviewer #1:

Remarks to the Author:

The manuscript by Ma et al. describes Citrus Huanglongbing as an immune-mediated disease where the effects of the disease are largely mediated by high accumulation of reactive oxygen species. The manuscript is very interesting and a significant contribution to the field. I am happy to state that the authors have addressed almost all my comments to my satisfaction.

Minor comment:

In Figure S10 the authors use the flavoprotein inhibitor DPI to prevent ROS accumulation by inhibiting NADPH oxidases. DPI leads to a decrease in ROS levels. However, while statistically significant it supports only a partial role of NADPH oxidases in the observed CLas-induced ROS production. I suggest that the statement "Intriguingly, ROS levels in CLas-positive stems were reduced by NADPH oxidase inhibitor diphenyleneiodonium (DPI) (fig. S10), supporting the notion that CLas triggers ROS production via RBOH genes." (Lines 331-333) is an overstatement and should be rephrased to suggest that RBOHs contribute to the ROS accumulation triggered by CLas but likely other components (peroxidases, ROS detoxification enzymes) likely contribute to this.

Reviewer #3:

Remarks to the Author:

[No further comments for authors]

Reviewer #4:

Remarks to the Author:

Having gone through the manuscript again and counter-checked the revisions, the authors have satisfactorily addressed my concerns.

The present manuscript is a much more improved version of the one that I reviewed earlier; the language and the scientific tone look okay.

I hold no reservation in recommending this manuscript for publication.

RESPONSE TO REVIEWERS' COMMENTS

REVIEWERS' COMMENTS

Reviewer #1 (Remarks to the Author):

The manuscript by Ma et al. describes Citrus Huanglongbing as an immune-mediated disease where the effects of the disease are largely mediated by high accumulation of reactive oxygen species. The manuscript is very interesting and a significant contribution to the field. I am happy to state that the authors have addressed almost all my comments to my satisfaction.

Response: We thank the reviewer for the encouraging comments.

Minor comment:

In Figure S10 the authors use the flavoprotein inhibitor DPI to prevent ROS accumulation by inhibiting NADPH oxidases. DPI leads to a decrease in ROS levels. However, while statistically significant it supports only a partial role of NADPH oxidases in the observed CLAs-induced ROS production. I suggest that the statement "Intriguingly, ROS levels in CLAs-positive stems were reduced by NADPH oxidase inhibitor diphenyleneiodonium (DPI) (fig. S10), supporting the notion that CLAs triggers ROS production via RBOH genes." (Lines 331-333) is an overstatement and should be rephrased to suggest that RBOHs contribute to the ROS accumulation triggered by CLAs but likely other components (peroxidases, ROS detoxification enzymes) likely contribute to this.

Response: We have revised the sentence as: "...supporting the notion that RBOHs contribute to the ROS accumulation triggered by CLAs, which probably also results from contributions from other components such as peroxidases."

Reviewer #4 (Remarks to the Author):

Having gone through the manuscript again and counter-checked the revisions, the authors have satisfactorily addressed my concerns.

The present manuscript is a much more improved version of the one that I reviewed earlier; the language and the scientific tone look okay.

I hold no reservation in recommending this manuscript for publication.

Response: We thank the reviewer for the positive comments.